# Dynamic effects of genetic variation on gene expression revealed following hypoxic stress in cardiomyocytes

Michelle C Ward[1,2†‡]*, Nicholas E Banovich[3,4†§], Abhishek Sarkar[3], Matthew Stephens[3,5], Yoav Gilad[1,3]*

[1]Department of Medicine, University of Chicago, Chicago, United States; [2]Department of Biochemistry and Molecular Biology, University of Texas Medical Branch, Galveston, United States; [3]Department of Human Genetics, University of Chicago, Chicago, United States; [4]Integrated Cancer Genomics Division, Translational Genomics Research Institute, Phoenix, United States; [5]Department of Statistics, University of Chicago, Chicago, United States

*For correspondence:
miward@utmb.edu (MCW);
gilad@uchicago.edu (YG)

[†]These authors contributed equally to this work

Present address: [‡]Department of Biochemistry and Molecular Biology, University of Texas Medical Branch, Galveston, United States; [§]Integrated Cancer Genomics Division, Translational Genomics Research Institute, Phoenix, United States

Competing interests: The authors declare that no competing interests exist.

**Abstract** One life-threatening outcome of cardiovascular disease is myocardial infarction, where cardiomyocytes are deprived of oxygen. To study inter-individual differences in response to hypoxia, we established an in vitro model of induced pluripotent stem cell-derived cardiomyocytes from 15 individuals. We measured gene expression levels, chromatin accessibility, and methylation levels in four culturing conditions that correspond to normoxia, hypoxia, and short- or long-term re-oxygenation. We characterized thousands of gene regulatory changes as the cells transition between conditions. Using available genotypes, we identified 1,573 genes with a *cis* expression quantitative locus (eQTL) in at least one condition, as well as 367 dynamic eQTLs, which are classified as eQTLs in at least one, but not in all conditions. A subset of genes with dynamic eQTLs is associated with complex traits and disease. Our data demonstrate how dynamic genetic effects on gene expression, which are likely relevant for disease, can be uncovered under stress.

## Introduction

Cardiovascular disease (CVD), which ultimately damages heart muscle, is a leading cause of death worldwide (*WHO, 2018*). CVD encompasses a range of pathologies including myocardial infarction (MI), where ischemia or a lack of oxygen delivery to energy-demanding cardiomyocytes results in cellular stress, irreparable damage, and cell death. Genome-wide association studies (GWAS) have identified hundreds of loci associated with coronary artery disease (*Nikpay et al., 2015*), MI, and heart failure (*Shah et al., 2020*), indicating the potential contribution of specific genetic variants to disease risk. Most disease-associated loci do not localize within coding regions of the genome, often making inference about the molecular mechanisms of disease challenging. That said, because most GWAS loci fall within non-coding regions, these variants are thought to have a role in regulating gene expression. One of the main goals of the Genotype-Tissue Expression (GTEx) project has been to bridge the gap between genotype and organismal level phenotypes by identifying associations between genetic variants and intermediate molecular level phenotypes such as gene expression levels (*GTEx Consortium et al., 2017*). The GTEx project has identified tens of thousands of expression quantitative trait loci (eQTLs); namely, variants that are associated with changes in gene expression levels, across dozens of tissues including ventricular and atrial samples from the heart. However, the eQTLs reported by GTEx explain a modest proportion of GWAS loci, and while increasing the diversity of tissues and sample sizes will enable further insight, orthogonal approaches also need to be considered.

It is becoming increasingly evident that many genetic variants that are not associated with gene expression levels at steady state, may be found to impact dynamic programs of gene expression in specific contexts. This includes specific developmental stages (*Cuomo et al., 2020*; *Strober et al., 2019*), or specific exposure to an environmental stimulus such as endoplasmic reticulum stress (*Dombroski et al., 2010*), hormone treatment (*Maranville et al., 2011*), radiation-induced cell death (*Smirnov et al., 2012*), vitamin D exposure (*Kariuki et al., 2016*), drug-induced cardiotoxicity (*Knowles et al., 2018*), and response to infection (*Alasoo et al., 2018*; *Barreiro et al., 2012*; *Çalışkan et al., 2015*; *Kim-Hellmuth et al., 2017*; *Manry et al., 2017*; *Nédélec et al., 2016*). The studies of context-specific dynamic eQTLs highlight the need to determine the effects of genetic variants in the relevant environment. Therefore, if we are to fully understand the effects of genetic variation on disease, we must assay disease-relevant cell types and disease-relevant perturbations. Most of the aforementioned studies were performed in whole blood or immune cells, which means that there are many cell types and disease-relevant states that have yet to be explored.

With advances in pluripotent stem cell technology, we can now generate otherwise largely inaccessible human cell types through directed differentiation of induced pluripotent stem cells (iPSCs) reprogrammed from easily accessible tissues such as fibroblasts or B-cells. One of the advantages of iPSC-derived cell types as a model system is that the environment can be controlled, and thus we can specifically test for genetic effects on molecular phenotypes in response to controlled perturbation. This is particularly useful for studies of complex diseases such as CVD, which result from a combination of both genetic and environmental factors.

The heart is a complex tissue consisting of multiple cell types, yet the bulk of the volume of the heart is comprised of cardiomyocytes (*Donovan et al., 2019*; *Pinto et al., 2016*), which are particularly susceptible to oxygen deprivation given their high metabolic activity. iPSC-derived cardiomyocytes (iPSC-CMs) have been shown to be a useful model for studying genetic effects on various cardiovascular traits and diseases, as well for studying gene regulation (*Banovich et al., 2018*; *Benaglio et al., 2019*; *Brodehl et al., 2019*; *Burridge et al., 2016*; *de la Roche et al., 2019*; *Ma et al., 2018*; *McDermott-Roe et al., 2019*; *Panopoulos et al., 2017*; *Pavlovic et al., 2018*; *Ward and Gilad, 2019*).

In humans, coronary artery disease can lead to MI (*Dzau et al., 2006*) which results in ischemia and a lack of oxygen delivery to energy-demanding cardiomyocytes. Given the inability of cardiomyocytes to regenerate, this cellular stress ultimately leads to tissue damage. Advances in treatment for MI, such as surgery to restore blood flow and oxygen to occluded arteries, have improved clinical outcomes. However, a rapid increase in oxygen levels post-MI can generate reactive oxygen species leading to ischemia-reperfusion (I/R) injury (*Giordano, 2005*). Both MI and I/R injury can thus ultimately influence the amount of damage in the heart. iPSC-CMs allow us to mimic the I/R injury process in vitro by manipulating the oxygen levels that cardiomyocytes are exposed to in vivo.

We thus designed a study aimed at developing an understanding of the genetic determinants of the response to a universal cellular stress, oxygen deprivation, in a disease-relevant cell type, mimicking a disease-relevant process. To do so, we established an in vitro model of oxygen deprivation (hypoxia) and re-oxygenation in a panel of iPSC-CMs from 15 genotyped individuals (*Banovich et al., 2018*). We collected data for three molecular level phenotypes: gene expression, chromatin accessibility, and DNA methylation to understand both the genetic and regulatory responses to this cellular stress. This framework allowed us to identify eQTLs that are not evident at steady state, and assess their association with complex traits and disease.

## Results

We differentiated iPSC-CMs from iPSCs of 15 Yoruba individuals that were part of the HapMap project (*Banovich et al., 2018*). To obtain a measure of variance associated with the differentiation process, and to more effectively account for batch effects, we replicated the iPSC-CM differentiation from three individuals three times, yielding 21 differentiation experiments in total. The proportion of iPSC-CMs in each cell culture was enriched by metabolic purification (see Materials and methods). On Day 20, cardiomyocyte differentiation cultures from each individual were split into sub-cultures for each of the four subsequent oxygen conditions and for assessment of cardiomyocyte purity (*Figure 1—figure supplement 1*). iPSC-CMs were matured by electrical pulsing and maintenance in cell culture for 30 days. On Day 30 (+/- 1 day), the median cardiomyocyte purity from one representative

sub-culture across individuals was 81% (40–97% range), determined by flow cytometry as the proportion of cells that were positive for the cardiac-specific marker, TNNT2 (*Figure 1—figure supplement 2*; *Supplementary file 1*; see Materials and methods).

We studied the response of the iPSC-CMs to hypoxia and re-oxygenation (*Figure 1A*). To do so, we first cultured the iPSC-CMs at oxygen levels that are close to physiological oxygen levels (10% oxygen - Condition A) for 7 days. We then subjected the iPSC-CMs to 6 hours of hypoxia (1% oxygen - Condition B), followed by re-oxygenation for 6 hours (10% oxygen - Condition C), or 24 hours (10% oxygen - Condition D) as previously described (*Ward and Gilad, 2019*). Oxygen levels were reproducibly controlled in cell culture (*Figure 1B*, *Supplementary file 1*). In order to determine whether the cardiomyocytes were affected by the changes in oxygen levels, we measured the enzymatic activity of released lactate dehydrogenase throughout the experiment, as a proxy for cytotoxicity. We also measured released BNP, a clinical marker of heart failure (*Maeda et al., 1998*). As expected, both cytotoxicity (p=0.01, *Figure 1—figure supplement 3A–B*) and BNP (p=$5\times10^{-6}$, *Figure 1—figure supplement 3C*) levels increased following hypoxia and long-term re-oxygenation.

With this system established, we sought to understand the contribution of the global gene regulatory response to the molecular and cellular response to hypoxia and re-oxygenation. To do so, we collected global gene expression data (using RNA-seq; n = 15 individuals), chromatin accessibility data (using ATAC-seq; n = 14), and DNA methylation data (using the EPIC arrays; n = 13; *Figure 1C*) in each condition. With these data we studied both the gene regulatory response to oxygen perturbation, as well as the interaction of the response with the underlying genotype of the assayed individuals.

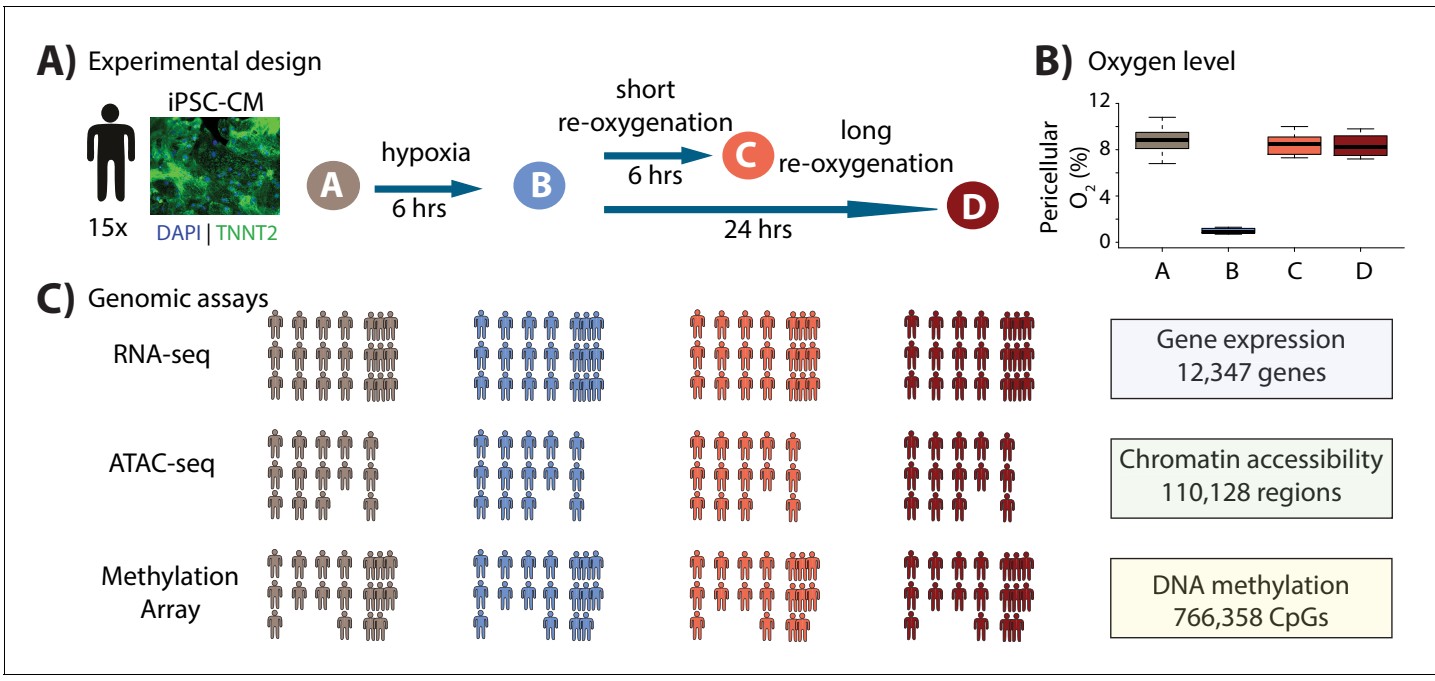

**Figure 1.** Study design to test the response of iPSC-derived cardiomyocytes to hypoxia and re-oxygenation. (**A**) Experimental design of the study. Cardiomyocytes differentiated from iPSCs (iPSC-CMs) from 15 Yoruba individuals were cultured in normoxic conditions (10% oxygen - condition A, brown) and subjected to 6 hr of hypoxia (1% oxygen - condition B, blue) followed by 6 and 24 hr of re-oxygenation (10% oxygen - conditions C [coral] and D [red]). Immunocytochemistry of a representative cardiomyocyte culture where green: TNNT2; blue: nuclei. (**B**) Peri-cellular oxygen levels of each condition. Each data point represents one individual undergoing the oxygen stress experiment. (**C**) Molecular phenotypes collected from each individual in each condition.

The online version of this article includes the following figure supplement(s) for figure 1:

**Figure supplement 1.** Experimental design.

**Figure supplement 2.** Purity of cardiomyocyte cultures.

**Figure supplement 3.** iPSC-CMs elicit a response to cellular stress.

## Gene expression changes in response to hypoxia and re-oxygenation

We first sought to identify those genes important for regulating the response by analyzing the gene expression (RNA-seq) data. We processed samples in batches as described in *Supplementary file 1* and mapped and filtered sequencing reads to prevent allelic mapping biases (*Figure 2—figure supplement 1*; *Supplementary file 1*; see Materials and methods; *van de Geijn et al., 2015*). We observed that one sample (18852A) was a clear outlier when comparing read counts for 18,226 autosomal genes across all samples, and thus excluded it from further analysis (*Figure 2—figure supplement 2*). We filtered out genes with low expression levels (see Materials and methods) to yield a final set with data from 12,347 expressed genes (see Materials and methods). We performed a number of correlation-based analyses using the data from the technical replicates (*Figure 2—figure supplement 3*), and confirmed that the quality of the data is high and that, in line with our flow cytometry data, our iPSC-CMs express a range of cardiomyocyte marker genes across conditions including *MYH7* and *TNNT2* (*Figure 2—figure supplement 4*).

We took advantage of the fact that we have replicate experiments from three individuals to correct the data for unwanted variation (see Materials and methods; *Risso et al., 2014*). Following this procedure, our samples clustered both by oxygen level and individual (*Figure 2—figure supplement 5*). To identify genes that respond to hypoxia and re-oxygenation, we first tested for differential expression between pairs of conditions using a linear model with a fixed effect for 'condition', a random effect for 'individual', and four unwanted factors of variation, learned from the data, as covariates. At an FDR of 10%, we identified thousands of genes that are differentially expressed between conditions (A vs. B = 4,983, B vs. C = 6,311; B vs. D = 6,792; A vs. D = 2,835; *Figure 2* and *Figure 2—figure supplement 6A*). In order to identify a single set of genes which respond to hypoxia across conditions, we used Cormotif (*Wei et al., 2015*) to jointly model pairs of tests. This approach led to the classification of 2,113 genes (17% of all expressed genes) as responding to hypoxia, which we term 'response genes' (*Figure 2*, *Figure 2—figure supplement 6B–C*), and 9,949 genes that do not change their expression level throughout the experiment which we term 'non-response genes'. Response genes are enriched for genes previously identified to respond to hypoxia in a Caucasian population of individuals (Chi-squared test; $p < 2.2 \times 10^{-16}$; *Ward and Gilad, 2019*), and are highly enriched for gene ontologies in transcription-related processes (see Materials and methods, modified Fisher's exact test; $p = 1 \times 10^{-19}$).

## Dynamic eQTLs are revealed following hypoxia

Having established that oxygen stress initiates a transcriptional response affecting thousands of genes, we sought to identify eQTLs, either before or after oxygen stress. Using the combined haplotype test (CHT), an approach that leverages allele-specific information in small sample sizes (see Materials and methods; *van de Geijn et al., 2015*), we identified 1,886 SNPs which associate with gene expression (eSNPs) in at least one condition resulting in 1,573 genes with eQTLs (eGenes) in at least one condition (q-value <0.1; A: 613; B: 564; C: 564 and D: 464; *Figure 3A–B*; *Supplementary file 2*). Given that cardiomyocytes were split from a single differentiation culture for each condition, we do not expect cell composition to bias the identification of eQTLs in each condition. Indeed, the distribution of eQTL effect sizes is similar across conditions (*Figure 3—figure supplement 1A*), and the eQTL effect size values are correlated across conditions (*Figure 3—figure supplement 1B*). In addition, we included the same number of individuals in each condition, RNA extraction batches included all conditions from a given individual, RIN scores are similar across conditions, and the number of sequencing reads are similar across conditions (*Supplementary file 1*). Together, these results suggest that our overall power to detect eQTLs is similar across conditions. We refer to the 613 eGenes identified in condition A as 'baseline eGenes'.

Our goal was to identify dynamic eQTLs, which are either revealed or suppressed as the cells transition between conditions. Due to the small sample size of our study, we have incomplete power to detect eQTLs in any condition; thus, a naive comparison of eQTLs classified as 'significant' across conditions will result in an over-estimation of the number of dynamic eQTLs. Indeed, the p-values of genes whose expression levels are not significantly associated with a SNP in any condition deviate from the expected uniform distribution (Kolmogorov–Smirnov test; $p < 2.2 \times 10^{-16}$; *Figure 3—figure supplement 2A*). To address this challenge, we first considered eQTLs identified using a q-value <0.1 in at least one condition, and visualized the p-value distributions of the corresponding eQTL

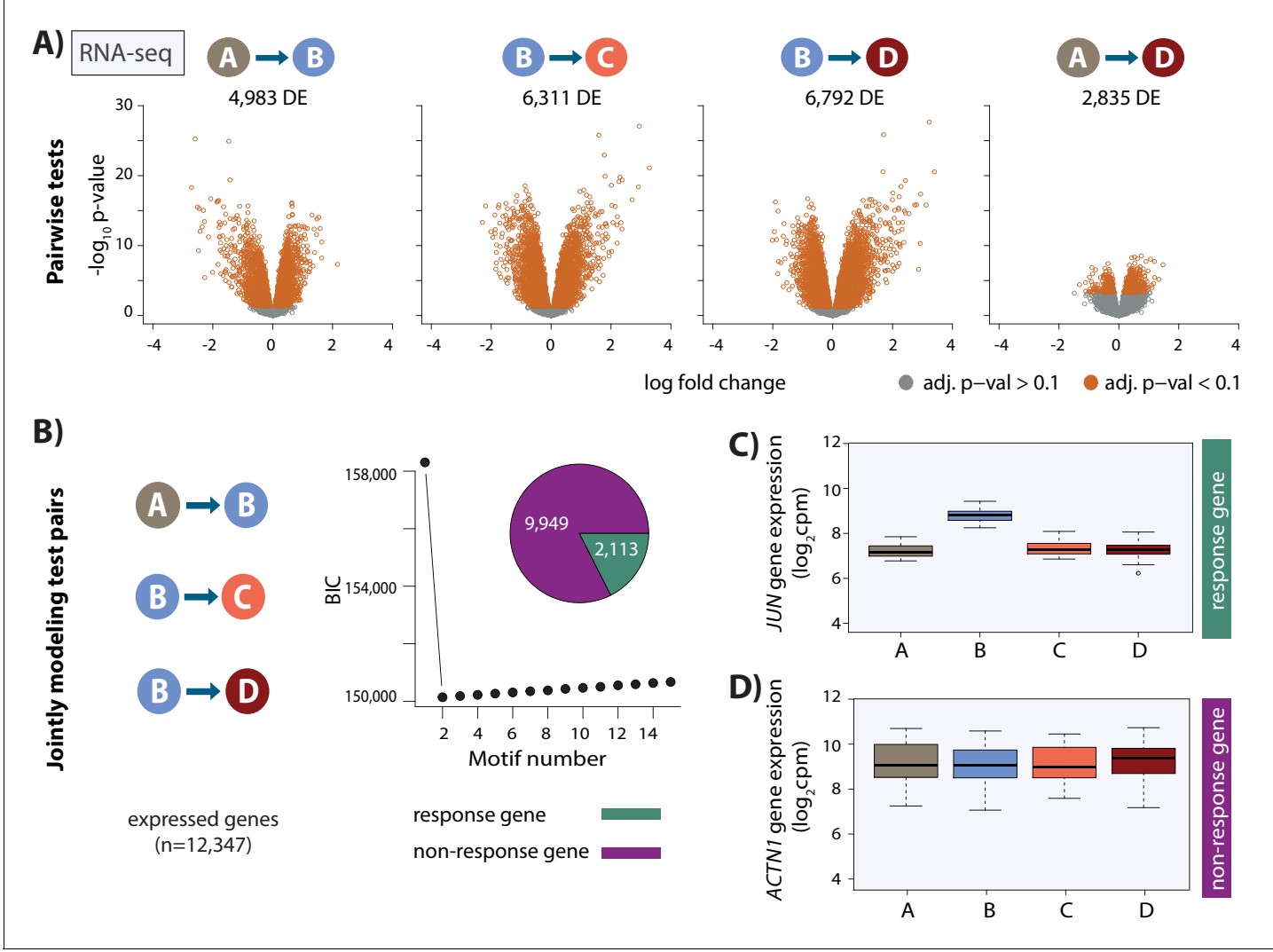

**Figure 2.** Thousands of gene expression changes occur following hypoxia and re-oxygenation. (A) Volcano plots representing genes that are differentially expressed (DE) between pairs of conditions (orange dots). (B) Bayesian information criterion (BIC) at increasing numbers of Cormotif correlation motifs following joint modeling of pairs of tests. Genes with a posterior probability of being differentially expressed p>0.5 across all tests are defined as 'response genes' and those with p<0.5 as 'non-response genes'. Inset shows the proportion of all expressed genes that are classified as response genes (green), and non-response genes (magenta). (C) Expression levels of *JUN*, a response gene, during the course of the experiment. (D) Expression levels of *ACTN1*, a non-response gene.

The online version of this article includes the following figure supplement(s) for figure 2:

**Figure supplement 1.** RNA integrity is similar across conditions.
**Figure supplement 2.** Correlation of read counts across samples.
**Figure supplement 3.** Correlation of gene expression measurements across samples.
**Figure supplement 4.** Cardiomyocyte marker genes are expressed in iPSC-CMs.
**Figure supplement 5.** RNA-seq samples cluster by oxygen level and individual.
**Figure supplement 6.** Hypoxia and re-oxygenation induces a gene expression response.

associations in all other conditions. These p-value distributions are expected to be uniform if we had complete power to detect eQTLs in any condition (because in that theoretical case, even a naive comparison of eQTLs classified as 'significant' across conditions will result in the identification of true condition-specific eQTLs). Due to incomplete power, this is clearly not the case (KS test; $p<2.2\times10^{-16}$; *Figure 3—figure supplement 2B*); however, this distribution allowed us to choose a lenient secondary p-value-based cutoff, where values deviate from the uniform distribution, to classify dynamic QTLs (p=0.15; *Figure 3—figure supplement 2B*).

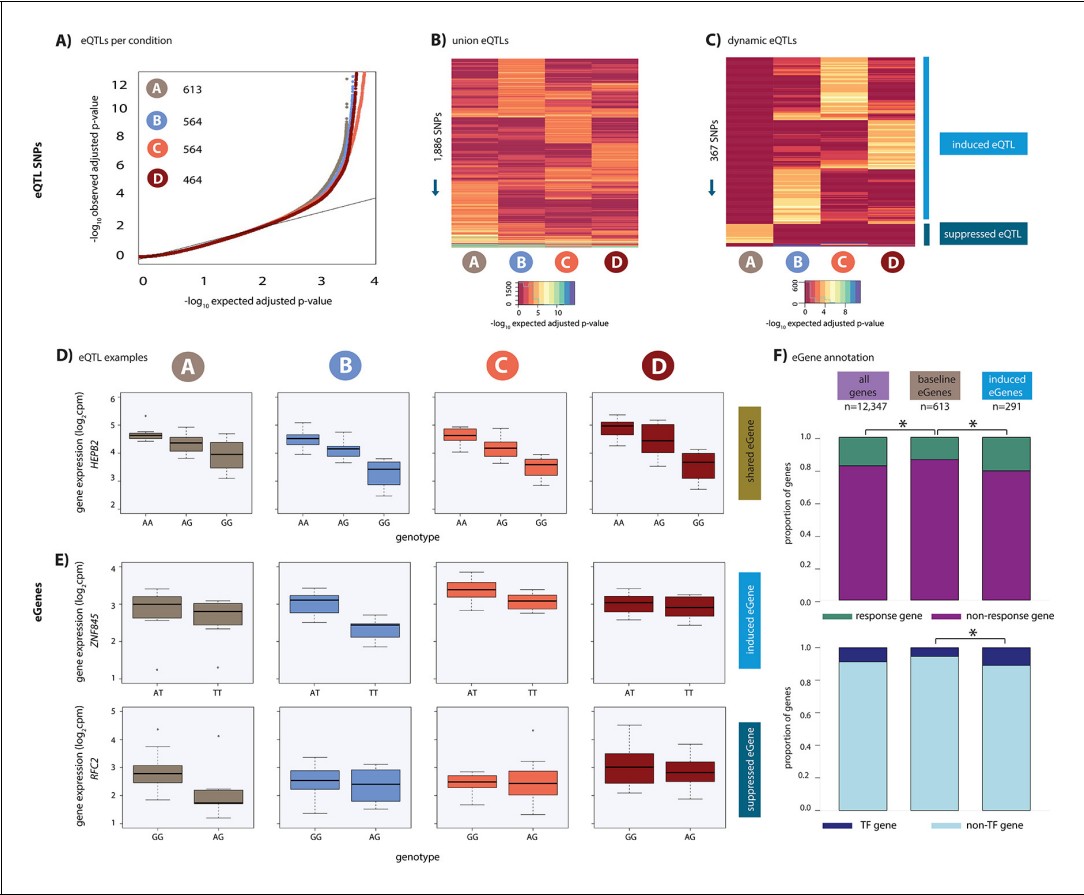

**Figure 3.** Hundreds of dynamic eQTLs are revealed following hypoxia and re-oxygenation. (**A**) QQ plot illustrating an enrichment of associations between genetic variants and gene expression levels in each condition. Numbers represent the number of eGenes in each condition. (**B**) Union of eQTLs identified in each condition. Each row represents a SNP that is an eQTL in at least one condition. Color represents the strength of the association p-value. (**C**) Heatmap illustrating the 367 SNPs that are classified as dynamic eQTLs including 330 induced eQTLs and 37 suppressed eQTLs. (**D**) An example of a shared eQTL, *HEPB2*. (**E**) Examples of each of the two dynamic eQTL categories. Top panel: genes that become an eQTL following hypoxia – induced eQTLs e.g. *ZNF845*. Bottom panel: genes that are an eQTL in normoxic condition A but not following hypoxia – suppressed eQTLs e.g. *RFC2*. (**F**) The proportion of all genes, baseline eGenes (all those identified in condition A), and induced eGenes that are also response genes, and transcription factor genes curated by Lambert et al. Asterisk denotes a statistically significant difference between gene categories (*p<0.05).

The online version of this article includes the following figure supplement(s) for figure 3:

**Figure supplement 1.** eQTL effect sizes in each condition.

**Figure supplement 2.** eQTL and dynamic eQTL identification.

**Figure supplement 3.** Dynamic eQTL examples.

We specifically focused on two dynamic scenarios. First, we defined suppressed eQTLs, as eQTLs that are identified in condition A at a q-value <0.1 but not in any of the other conditions, with a p-value greater than 0.15 (37 instances; *Figure 3C*). Second, we defined induced eQTLs as eQTLs identified in conditions B, C, or D at a q-value <0.1, but not in A, with a p-value greater than 0.15 (330 instances; *Figure 3C*). This set of 367 dynamic eQTLs corresponds to 328 unique dynamic eGenes (see Materials and methods) and includes *ZNF845* and *RFC2* (*Figure 3E*) as well as *PPARGC*1, *C8orf82* and *CELF1* (*Figure 3—figure supplement 3*). While our choice of the particular statistical cutoffs is somewhat arbitrary, we can evaluate the false discovery rate associated with our chosen cutoff. Based on the p-value distributions of the corresponding eQTL associations in all other conditions, we estimate that our approach to classify dynamic eQTLs is associated with a false discovery rate of 48%. The relatively high FDR associated with our choice of statistical cutoffs does not indicate that these loci are not eQTLs; rather it means that if we had a larger sample size, roughly

half of our dynamic eQTLs should have been classified as eQTLs in more conditions, potentially in all of them. As expected, using a more stringent p-value cutoff of 0.9, we find fewer dynamic eQTLs (19) and a lower false discovery rate of 7%.

We next wanted to determine whether the dynamic eQTLs we identified in iPSC-CMs are also eQTLs in primary heart tissue. To do so, we compared our 367 dynamic eSNPs and eSNPs regulating the same gene in all four conditions ('shared eQTLs', n = 20) to eQTLs identified in left ventricle heart tissue and 49 tissues assayed from hundreds of individuals in the GTEx study (*GTEx Consortium et al., 2017*). Thirty-two of 326 dynamic eSNP-eGene pairs tested in GTEx are eQTLs in heart tissue (9.8%), and 140 are eQTLs in at least one other assayed tissue (42.9%) demonstrating that many of these genes have context-dependent inter-individual variation in expression. Six of 19 shared eSNP-eGene pairs tested in GTEx are eQTLs in heart tissue (31.6%), and six are an eQTL in at least one other tissue (31.6%). Shared eSNPs are therefore more likely to be eSNPs in heart tissue than dynamic eSNPs (Chi-squared test; p=0.01). To further investigate eQTL concordance, we compared the eQTL effect size of our dynamic and shared eQTLs to the eQTL effect size in heart tissue determined by the GTEx consortium. Shared eQTLs overlapping heart tissue eQTLs (n = 6) have a higher concordance of effect than dynamic eQTLs overlapping heart tissue eQTLs (n = 32; Spearman correlation = 0.73 vs. 0.12) suggesting that our perturbation study has revealed novel eQTL effects.

## Dynamic eGenes are enriched for response genes and transcription factors

To determine whether induced eGenes coincide with expression changes of the same genes following hypoxia, we integrated the results of our eQTL and differential expression analyses. For this analysis, we compared eGenes identified in condition A, baseline eGenes, with the set of induced eGenes. In line with our previous findings, baseline eGenes, as well as left ventricle (LV) and atrial appendage (AA) eGenes found in primary heart tissue (GTEx), are depleted for response genes (Chi-squared test; p<0.02, *Ward and Gilad, 2019*). However, we found a significant enrichment in response genes amongst induced eGenes (57 of 277 genes; Chi-squared test; p=0.01, *Figure 3F*) when compared to baseline eGenes, suggesting that induced eQTLs often impact the regulation of genes that respond to hypoxic stress.

Given that thousands of genes are differentially expressed in response to hypoxia, and many of these genes correspond to transcription-related processes, we next investigated the role of 1,639 genes annotated as transcription factors in humans, which may drive the transcriptome changes we observe (*Lambert et al., 2018*). We found a significant enrichment of genes annotated as transcription factors amongst the genes responding to oxygen stress compared to non-response genes (327 of 1,639 annotated human TFs, chi-squared test; $p<2\times10^{-16}$; *Figure 3F*). Given that stress affects transcription factor gene expression, we asked whether induced eGenes are also enriched for transcription factor genes. Indeed, genes annotated as transcription factors are enriched in induced eGenes compared to baseline eGenes (32 TFs; p=0.004), including *MITF* and *PPARA*, both of which are TFs that have been previously implicated in hypoxic response (*Feige et al., 2011*; *Narravula and Colgan, 2001*). TFs amongst induced eGenes are more likely to be response genes than non-response genes (p=0.02); however, there is no difference in the proportion of response genes amongst baseline eGenes annotated as TFs and induced eGenes annotated as TFs (38% vs. 39%). The enrichment we observe in TFs amongst our induced eGenes suggests that latent genetic variation can have multiple downstream effects on gene expression including gene targets of TF genes. This could provide a mechanism for the appearance of induced eGenes that are neither response genes nor TF genes.

## Chromatin accessibility changes following hypoxia and re-oxygenation

We next asked whether the hundreds of transcription factor expression changes following hypoxic stress are accompanied by global chromatin accessibility changes. To examine this, we performed ATAC-seq experiments in all four conditions to identify regions of open chromatin (we were only able to collect these data from 14 of the 15 individuals; *Figure 1—figure supplement 1*, see Materials and methods). We filtered the ATAC-seq reads to include only those reads that map unambiguously to the nuclear genome (see Materials and methods, *Figure 4—figure supplement*

1). We identified a set of open chromatin regions in each sample, and merged samples across individuals within each condition. Genomic regions identified as accessible in each condition were then merged to yield a set of 128,672 open chromatin regions across conditions (with a median length of 312 bp). Regions with low read counts were filtered out, resulting in a final set of 110,128 regions. Analysis of various metrics revealed the data to be of good quality (*Figure 4—figure supplements 2–3*).

We sought to identify chromatin regions that are differentially accessible across pairs of conditions. Using a sensitive adaptive shrinkage-based approach with a False Sign Rate of 10% (*Stephens, 2017*), we could not detect changes in accessibility between baseline and hypoxia across 110,128 regions; however, we identified 831 differentially accessible regions (DARs) between hypoxia and short-term re-oxygenation (BC-DARs; 429 regions with increased accessibility and 402 with decreased accessibility), and 71 DARs between hypoxia and long-term re-oxygenation (BD-DARs; *Figure 4A*). Despite the fact that we do not identify any DARs between normoxia and hypoxia, there is a strong anti-correlation in the effect size between the normoxia (A) and hypoxia (B) conditions, and the hypoxia (B) and re-oxygenation (C) conditions across regions (Spearman correlation = −0.62; sign test p=$4.6\times10^{-14}$; *Figure 4B*) suggesting minor changes in accessibility in response to hypoxia that do not meet our significance threshold. Conversely, there is a strong correlation in effect sizes between hypoxia and short-term re-oxygenation (BC-DARs), and hypoxia and long-term re-oxygenation (BD-DARs; Spearman correlation = 0.74; *Figure 4C*), and 59 of the 71 BD-DARs are amongst the 831 BC-DARs (83%), suggesting that most regions have returned to baseline levels of accessibility by the first re-oxygenation condition. This includes a region within the intron of the *FOXO1* gene, a master regulator of the oxidative stress response (*Figure 4D*). We therefore considered the 831 BC-DARs, henceforth DARs, in further analysis. These DARs represent 0.8% of the total number of accessible regions.

## Linking chromatin accessibility changes with gene expression changes

We next sought to integrate our gene expression data with our chromatin accessibility data. We found that when considering a 50 kb window around the TSS of expressed genes, DARs are enriched near response genes compared to non-response genes (Chi-squared test; p=0.03). Of 2,113 response genes, 113 have a DAR within 50 kb of the TSS. This set includes an accessible region, overlapping a HIF1α site, within 500 bp of the 3' end of the classic hypoxia response gene, *ADM* (*Figure 4E*).

We asked whether the changes in chromatin accessibility coincide with the appearance of dynamic eQTLs. We found that DARs are no more likely to be near dynamic eGenes than shared or baseline eGenes. In line with previous estimates of the proportion of eQTLs in open chromatin regions, 24 baseline eQTL SNPs (613 total SNPs) and 19 dynamic eQTL SNPs (367 total SNPs) overlap with accessible chromatin regions (*GTEx Consortium et al., 2017*). One dynamic eQTL SNP overlaps a DAR, near the actin filament binding protein gene, *FGD4*. This gene was also shown to be differentially expressed between children with congenital heart defects where the defect leads to a chronic hypoxic state (cyanotic disease), and children with a similar defect but where oxygen levels are not affected (acyanotic disease; *Ghorbel et al., 2010*).

Genetic variants within transcription factor binding sites can influence chromatin accessibility at these regions, leading to effects on gene expression. To directly test whether there are genetic effects on chromatin accessibility, independent of gene expression, we sought to identify chromatin accessibility QTLs (caQTLs) that is genetic variants located within the 128,672 accessible regions, which coincide with different levels of accessibility based on genotype. We identified few caQTLs per condition (q-value <0.1; A: 10, B: 1, C: 7, D: 6; *Figure 4—figure supplement 4A*). Six of these caQTLs are classified as dynamic caQTLs that is induced or suppressed in response to hypoxia using the same definitions as used for the dynamic eQTLs, and include regions at the TSS of the mRNA decapping enzyme gene *DCPS*, and a region within 100 kb of the *C1orf99* gene (*Figure 4—figure supplement 4B–C*).

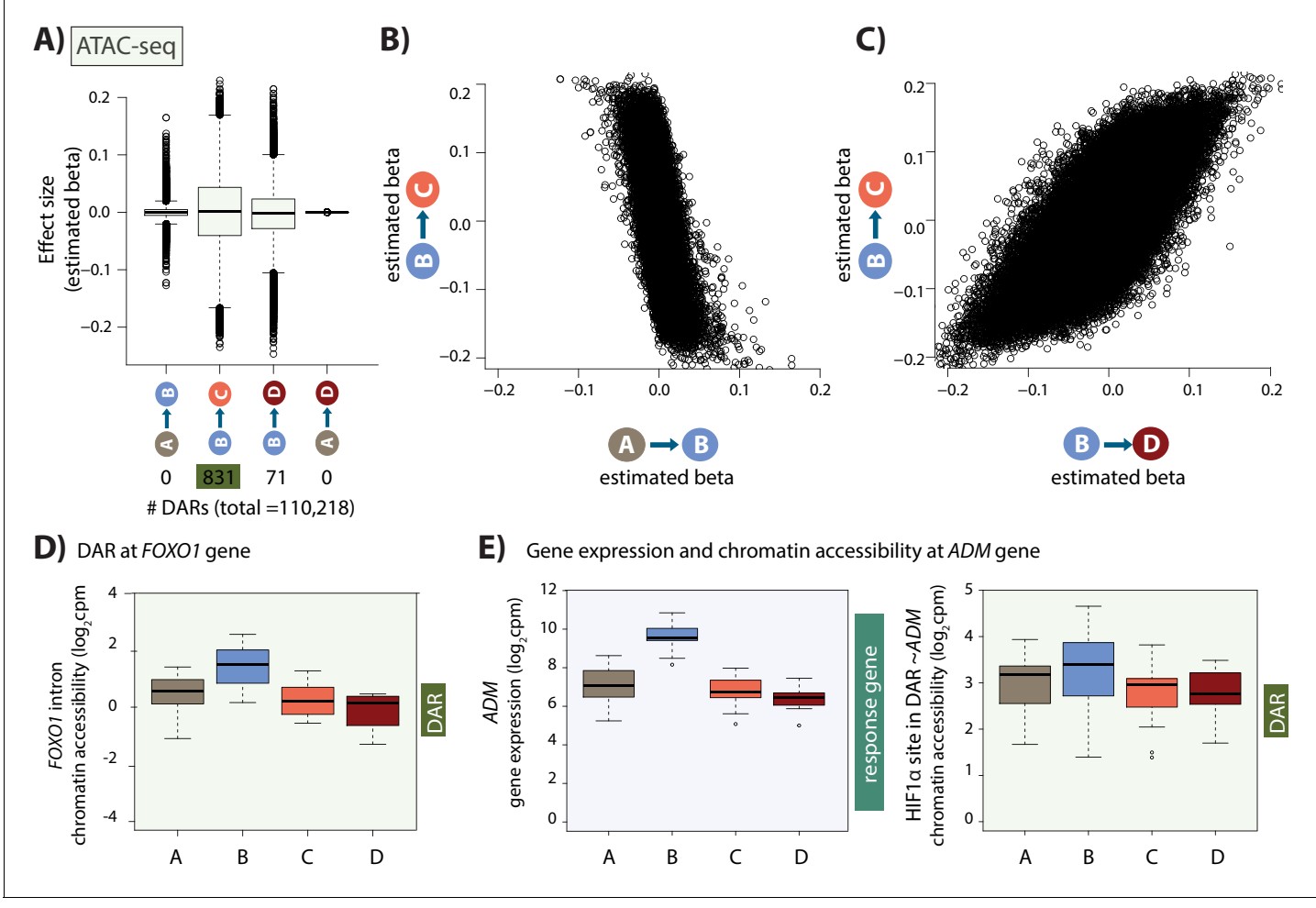

**Figure 4.** Hundreds of chromatin accessibility changes occur following hypoxia and re-oxygenation. (A) Estimated effect size (Beta) following adaptive shrinkage (ash) between each pair of conditions, and the numbers of chromatin regions that are differentially accessible (DARs) between pairs of conditions. (B) Estimated Beta for each region when comparing A vs. B and B vs. C. (C) Estimated Beta for each region when comparing B vs. C, and B vs. D. (D) Chromatin accessibility levels at a chromatin region within a *FOXO1* intron. (E) Expression levels of the hypoxia-responsive gene *ADM* following hypoxia, and chromatin accessibility levels at a DAR, overlapping an induced HIF1α-bound region, within 500 bp of the *ADM* gene. The online version of this article includes the following figure supplement(s) for figure 4:

**Figure supplement 1.** Numbers of ATAC-seq reads are similar across conditions.
**Figure supplement 2.** ATAC-seq library quality control.
**Figure supplement 3.** ATAC-seq libraries cluster by individual and treatment.
**Figure supplement 4.** Identification of caQTLs.

## Genomic features associated with differentially accessible regions (DARs)

We next wanted to determine what distinguishes DARs from constitutively accessible regions (CARs). To do so, we investigated three classes of genomic features: (1) promoter- and enhancer-associated marks, (2) transcription factor binding locations, and (3) underlying DNA sequence features. We found that DARs are more likely to overlap TSS than constitutively accessible regions (43% overlap vs. 11% overlap; chi-squared test, $p<2\times10^{-16}$; *Figure 5A*) suggesting that DARs may be involved in the gene regulatory response. Indeed, DARs are more likely to coincide with active histone marks in left ventricle heart tissue than constitutively accessible regions (H3K4me3: 57% overlap DARs vs. 24% overlap constitutively accessible regions; chi-squared test; $p<2.2\times10^{-16}$; H3K4me1: 86% overlap DARs vs. 52% overlap constitutively accessible regions; $p<2.2\times10^{-16}$; *Figure 5B*).

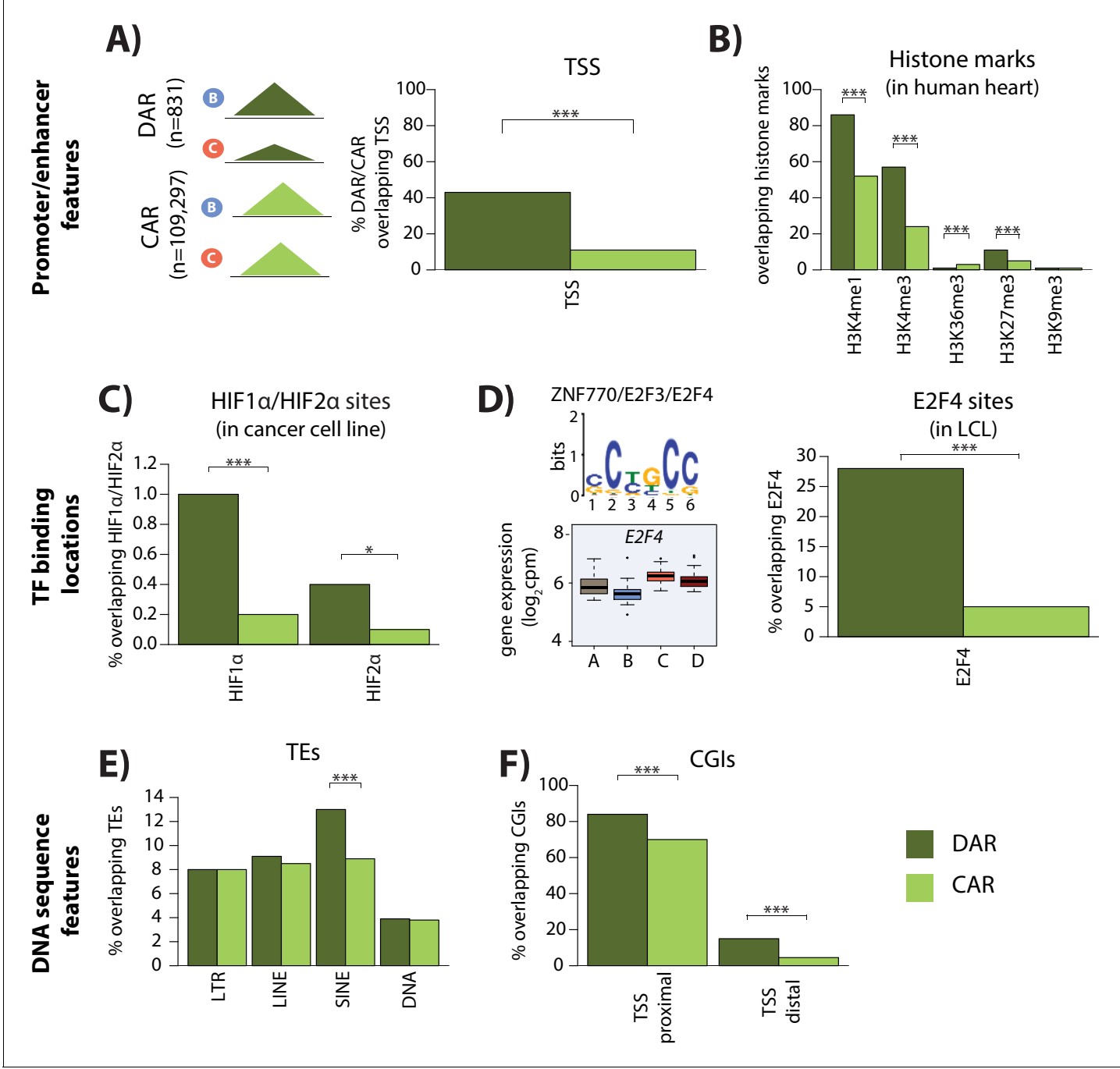

**Figure 5.** Differentially accessible regions are enriched for active chromatin features. (**A**) The proportion of differentially accessible regions (DARs) and constitutively accessible regions (CARs) that overlap with annotated TSS. (**B**) The proportion of DARs and CARs that overlap with the locations of histone marks determined by ChIP-seq in human heart tissue (*ENCODE Project Consortium, 2012*). (**C**) The proportion of DARs and CARs that overlap with HIF1α and HIF2α binding locations determined by ChIP-seq in a breast cancer cell line (*Schödel et al., 2011*). (**D**) The most significant motif identified to be differentially enriched in DARs compared to all ARs that is putatively recognized by ZNF770, E2F3 and E2F4. We classify *E2F4* as a response gene and therefore determined the proportion of DARs and CARs that overlap with E2F4 ChIP-seq binding locations identified in a human LCL line (*Lee et al., 2011*). (**E**) The proportion of DARs and CARs that overlap four major transposable element (TE) classes – LTR, LINE, SINE, DNA. The proportion of DARs and CARs that overlap with CpG islands (CGIs) that is proximal to the TSS (+/- 2 kb from the TSS), and distal to the TSS. Asterisk denotes a statistically significant difference between DARs and CARs (*p<0.05, **p<0.005, ***p<0.0005).

To determine whether sequence-specific hypoxia-responsive transcription factors associate with differentially accessible chromatin, we integrated DARs with published chromatin immunoprecipitation followed by high-throughput sequencing (ChIP-seq) data for the well-studied hypoxia-inducible factors HIF1α and HIF2α (*Schödel et al., 2011*). A total of 234 of the 356 HIF1α ChIP-seq binding sites (66%), and 150 of the 301 HIF2α binding sites (50%) overlap with our set of 110,128 accessible chromatin regions. We found that these HIF1α and HIF2α binding sites are more likely to overlap the 831 differentially accessible regions than the 109,275 constitutively accessible regions (Fisher test; p=0.03; *Figure 5C*).

We next took an unbiased approach to identify transcription factor binding motifs that are enriched in DARs compared to all accessible regions using MEME-ChIP software (see Materials and methods). We found two motifs to be enriched in DARs compared to all regions (*Figure 5D*). Motif 1 ($p=2\times10^{-2}$) is recognized by *HTF4* and *TFE2,* both of which are non-response genes in our system. Motif 2 ($p=6.2\times10^{-42}$) is posited to be recognized by ZN770, E2F3, and E2F4. Both *ZN770* and *E2F4* are response genes in our system. DARs arise between the hypoxia and re-oxygenation conditions, and *E2F4* expression increases following re-oxygenation, suggesting that it may be involved in the response. To test this hypothesis, we obtained a published ChIP-seq data set for E2F4 (*Lee et al., 2011*), and overlapped the 16,245 E2F4-bound regions with DARs and constitutively accessible regions. E2F4-bound regions are significantly enriched in DARs compared to constitutively accessible regions (Chi-squared test; $p<2.2\times10^{-16}$; *Figure 5D*). E2F4 is important for survival following ischemia in neurons, and has been suggested to be an anti-apoptotic factor in cardiomyocytes (*Dingar et al., 2012*; *Iyirhiaro et al., 2014*).

To identify additional sequence features that associate with DARs, we asked whether transposable elements (TE), a potential source of regulatory sequence subjected to chromatin-level regulation, are enriched in these sites (*Du et al., 2016*; *He et al., 2019*). We found that while three of the main TE classes, LINEs, LTRs, and DNA elements are similarly enriched in DARs compared to constitutively accessible regions; SINEs are specifically enriched in DARs ($p=6.5\times10^{-6}$; *Figure 5E*). There is an enrichment of both Alu and MIR SINE family members in DARs ($p=3\times10^{-5}$ and p=0.006). AluS elements, and the AluSq and AluSp sub-families, are particularly enriched within the Alu family (p=0.007 and p=0.02 respectively). A different cellular stress, heat shock, has previously been shown to remodel chromatin accessibility at Alu elements in cervical cancer cells (*Kim et al., 2001*).

As Alu and MIR TE sequences are notably CpG dense (*Medstrand et al., 2002*), we next asked about the enrichment of CpG-dense CpG islands (CGIs) in our differentially accessible regions. We found that CpG islands are enriched in DARs compared to constitutively accessible regions, whether these regions fall within 1 kb of TSS, which are typically enriched for CGIs, or not ($p<2.2\times10^{-16}$; *Figure 5F*).

## DNA methylation state at stress-responsive genes and chromatin regions

Genes with CGI promoters are thought to allow flexibility in TSS choice compared to genes without CGI promoters (*Carninci et al., 2006*), and to allow for the rapid induction of gene expression in response to stimuli (*Ramirez-Carrozzi et al., 2009*). We therefore asked whether this promoter feature is enriched in the stress response genes. Indeed, we find that response genes are more likely to have CGI promoters than non-response genes (Chi-squared test; p=0.002).

Given the enrichment of CpG islands in gene promoters and chromatin regions that are responsive to stress, we asked whether this feature corresponded to differences in CpG DNA methylation levels in these same regions. We measured global DNA methylation levels at 766,658 CpG sites in all conditions from 13 of our individuals (no data was collected from two of the 15 individuals), together with 23 replicate samples from three individuals (see Materials and methods; *Supplementary file 1*). We found the expected bimodal distribution of DNA methylation Beta-values across CpGs (Beta-values represent the ratio of intensities between the methylated and unmethylated alleles; *Figure 6—figure supplement 1A*). Additional analyses indicated the data to be of good quality (*Figure 6—figure supplements 1–2*).

To determine whether steady-state DNA methylation levels mark genes or regions that will change their expression level in response to stress, we investigated baseline DNA methylation levels in the promoters of genes classified as response genes and non-response genes, as well as accessible regions and DARs. To do so, we assessed the DNA methylation level at CpGs within 200 bp

upstream of the TSS in the baseline condition (condition A). The majority of the assayed CpGs were hypomethylated with a median Beta-value of less than 0.2 across genes and regions (*Figure 6A*). While there is no difference in median DNA methylation levels between response and non-response genes, we found that the median DNA methylation level is lower in DARs compared to all accessible regions ($p < 2.2 \times 10^{-16}$). These data suggest that responsive chromatin regions may have specific epigenetic profiles which poise them for rapid response to stress.

## DNA methylation levels are largely stable following hypoxia and re-oxygenation

Given that DNA methylation levels can associate with gene expression levels, we asked whether any CpGs are differentially methylated during the course of the oxygen perturbation experiment, which induces thousands of gene expression changes. When considering all 766,658 CpGs we did not find any differentially methylated CpGs across any pair of conditions (10% FDR; *Figure 6B*), and all p-values are estimated to be true null p-values (pi0 = 1 when estimated by q-value across all pairs of conditions; *Figure 6—figure supplement 3*). We were also unable to detect any differentially methylated CpGs when considering two estimates of DNA methylation levels: Beta-values or M-values ($\log_2$ ratio of intensities of methylated versus unmethylated alleles).

To increase the likelihood of identifying differentially methylated CpGs, we reduced the number of statistical tests for identifying differentially methylated CpGs by restricting our test set of CpGs to those within annotated regions of interest. Because of the CpG island enrichment in our response gene promoters, we first selected CpGs present within CpG islands (143,587 of the 766,658 measured CpGs). Again, we found no differentially methylated CpGs between baseline (A) and hypoxia (B), and hypoxia (B) and the short-term re-oxygenation condition (C). However, we identified four differentially methylated CpGs (DMCpGs) between the baseline (A) and long-term re-oxygenation condition (D; *Figure 6C*). This set includes a CpG in the intron of the *EGR2* response gene, which shows increased DNA methylation levels over time (*Figure 6D*). Methylation at CpG islands within the intron of *EGR2* has been shown to confer enhancer activity in cancer cells (*Unoki and Nakamura, 2003*). If we only select CpGs located within the promoters of the 2,113 response genes, we find one DMCpG within the promoter of the *FTSJ2* gene, a rRNA methyltransferase, that is differentially methylated between the hypoxia (B) and long-term re-oxygenation (D) conditions. Selecting CpGs located only within the 831 DARs reveals two DMCpGs between baseline (A) and hypoxia (B), and one DMCpG between baseline (A) and long-term re-oxygenation (D). We therefore were only able to identify a handful of differentially methylated CpGs during our experimental timecourse which elicited thousands of gene expression changes.

To further investigate the apparent differences in response to hypoxia across our three molecular phenotypes, we calculated the proportion of variance explained by 'individual', 'condition' and 'replicate'. We observed a lower proportion of variance explained by 'condition' in the DNA methylation data (0.8%) and chromatin accessibility data (3%) compared to the gene expression data (6%, t-test, $p < 2.2 \times 10^{-16}$), and a corresponding increase in the residual variance (*Figure 6—figure supplement 4*). For the DNA methylation data in particular, there is a similar proportion of variance explained by individual (44% for gene expression, 40% for DNA methylation, and 28% for chromatin accessibility) and approximately an order of magnitude less variance explained by condition. These results suggest that noise alone cannot explain the relatively small number of chromatin accessibility and DNA methylation changes.

## Dynamic eQTLs associate with traits and disease

Finally, we wanted to determine whether any of the dynamic eQTL SNPs or genes that we identified might also be associated with complex traits or disease. To maximize the number of potentially phenotypically-relevant genomic loci, we performed three independent analyses. We first took an unbiased, SNP-based approach by searching within a catalog of genetic variants associated with a variety of traits assayed in thousands of GWAS for overlap with our dynamic eSNPs (*Buniello et al., 2019*). By intersecting these two data sets, we found one induced dynamic eSNP (rs8053350) that is also associated with a measured phenotype in GWAS – varicose veins (*Supplementary file 1*; *Figure 7—figure supplement 1A-B*).

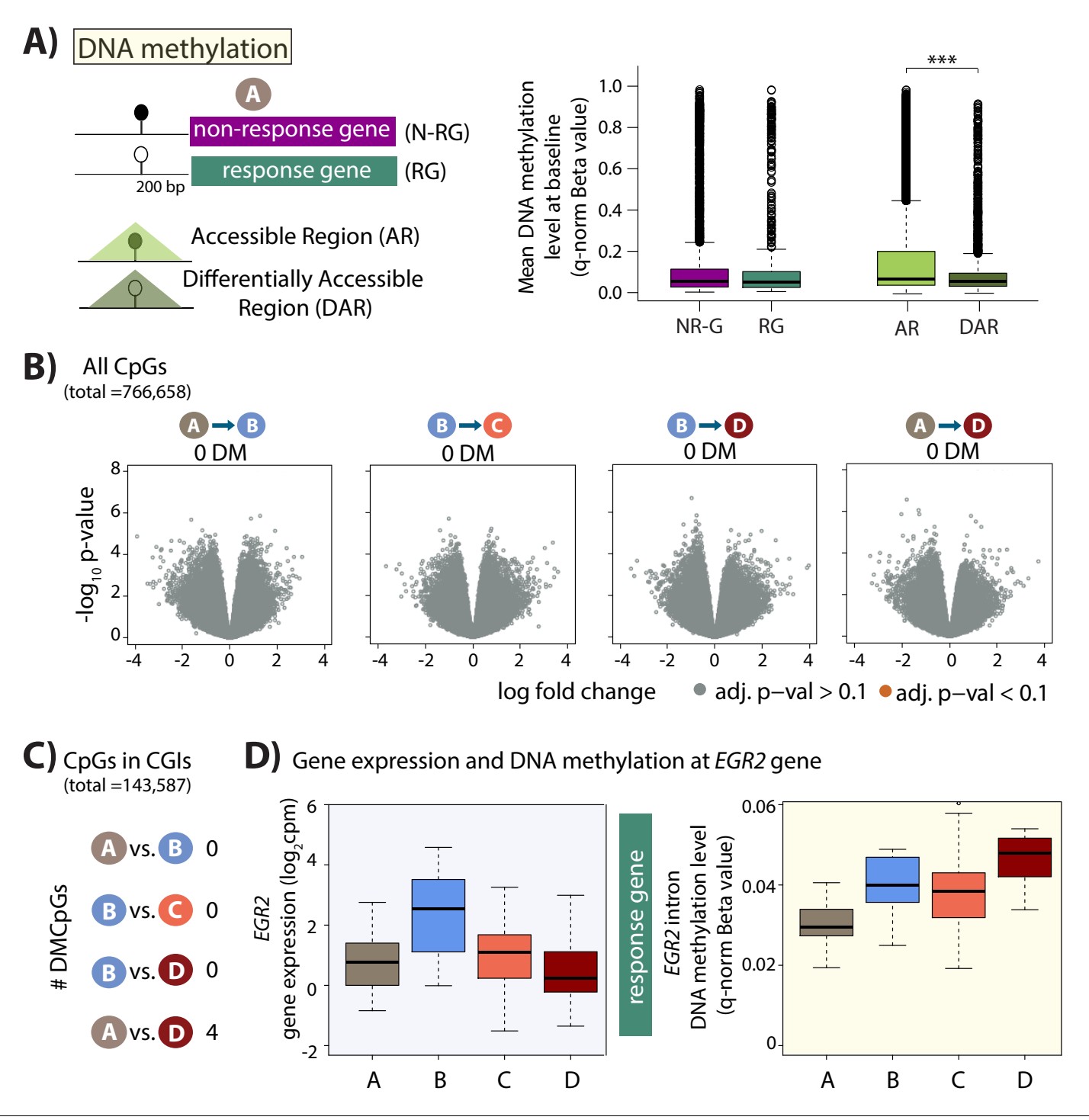

**Figure 6.** Minimal DNA methylation changes following hypoxia. (**A**) Mean DNA methylation levels (Beta-values) in the baseline condition (A) at CpGs within 200 bp upstream of the TSS of non-response genes (NR-G) and response genes (RG), and within all accessible regions (ARs) and differentially accessible regions (DARs; *p<0.05, ***p<0.0005). (**B**) Volcano plots representing the differential DNA methylation analysis across 766,658 CpGs. There are no differentially methylated (DM) CpGs across any pair of conditions. (**C**) Numbers of differentially methylated CpGs, when restricting the test set to include only CpGs within CpG islands, across pairs of conditions. (**D**) Gene expression and DNA methylation levels at a differentially methylated CpG within an intron of the *EGR2* response gene.

The online version of this article includes the following figure supplement(s) for figure 6:

**Figure supplement 1.** DNA methylation array quality control.

*Figure 6 continued on next page*

**Figure supplement 4.** Variance associated with oxygen condition is lower in DNA methylation data compared to gene expression data.

Given that our eQTL data are from the Yoruba population from West Africa and that most GWAS studies are performed in European populations, which have an inherently different LD structure, we may not expect to see strong overlap at the level of individual variants. We therefore next took an orthogonal gene-based approach, using the same GWAS catalog, to specifically investigate the genes implicated in three phenotypes associated with cardiovascular function or response to oxygen deprivation: MI, heart failure, and stroke (see Materials and methods). If the expression of these CVD genes is found to vary across individuals following hypoxia, it suggests that they may be important for mediating disease risk. Indeed, we found that six of our dynamic eGenes are implicated in these three disease states by GWAS gene mapping approaches (*Supplementary file 1*). This list includes the DNA damage and apoptosis factor *ZC3HC1,* which is implicated in MI and stroke (*Figure 7—figure supplement 1C–D*). Importantly, *ZC3HC1* is not an eGene in LV or AA, but the SNP-gene pair is an eQTL in other tissues.

Lastly, we performed an in-depth SNP-based analysis using full summary statistics for coronary artery disease (CAD) and MI from the CARDIoGRAMplusC4D Consortium (*Stitziel et al., 2016*; *Nikpay et al., 2015*; *Webb et al., 2017*). We tested whether the lead eSNP from our set of dynamic eQTLs is also associated with CAD or MI and identified two loci (*Supplementary file 1*). This set includes the eSNP (rs12588981) for the dynamic eQTL *EIF2B2* which is nominally associated with CAD (GWAS association $p=4.7\times10^{-5}$) and is in high and statistically significant LD with the lead GWAS variant (GWAS association $p=9.9\times10^{-8}$) at this locus (rs3832966; $R^2 = 0.96$, p<0.001; *Figure 7*; *Figure 7—figure supplement 2*). Given that we have incomplete power to detect dynamic eQTLs, we also used the same GWAS dataset to investigate whether any of the significant eQTL SNPs in any condition are associated with CAD or MI. We determined that eSNP (rs8105092), which regulates *CARM1* in the hypoxic condition (condition B), is nominally associated with MI (GWAS association $p=1.79\times10^{-5}$) and in moderate, yet significant LD, with the lead GWAS variant (GWAS association $p=2.8\times10^{-9}$) at this locus (rs4804142; $R^2 = 0.32$, p<0.001; *Figure 7—figure supplement 3*), and is not an eGene in LV or AA. A handful of the eQTLs that we highlight here were tested for association with gene expression by the GTEx consortium, yet are not eQTLs in heart tissue. These results suggest that perturbation studies in relevant cell types can give insight into the molecular basis for the genetic association with complex traits and disease, which might not be gleaned from the study of post-mortem tissues.

## Discussion

Studying gene expression across individuals in response to stress can reveal latent effects of genetic variation, which may contribute to higher-order phenotypes and disease. In order to understand the effects of genetic variation in a disease-relevant cell type and a disease-relevant process, we differentiated cardiomyocytes from a panel of genotyped individuals, and subjected them to hypoxia and re-oxygenation. We found hundreds of eQTLs that are revealed or suppressed following hypoxic stress (dynamic eQTLs), several of which have been associated with phenotypes measured in GWAS.

### Steady-state and dynamic eQTLs may help understand CVD

Attempts have been made to identify genetic variants that associate with gene expression levels and CVD phenotypes in easily accessible biological samples such as blood. However, less than half of CVD/MI GWAS loci are associated with an eQTL in whole blood when thousands of individuals are tested (*Joehanes et al., 2017*). To determine the effects of genetic variation on gene expression specifically in the heart, more targeted studies have taken advantage of left ventricle tissue (*GTEx Consortium et al., 2017*; *Koopmann et al., 2014*), left atrium tissue (*Lin et al., 2014*; *Sigurdsson et al., 2017*), and right atrial appendage tissue (*GTEx Consortium et al., 2017*) obtained during cardiac surgeries or post-mortem. Using fewer than a hundred individuals, a handful

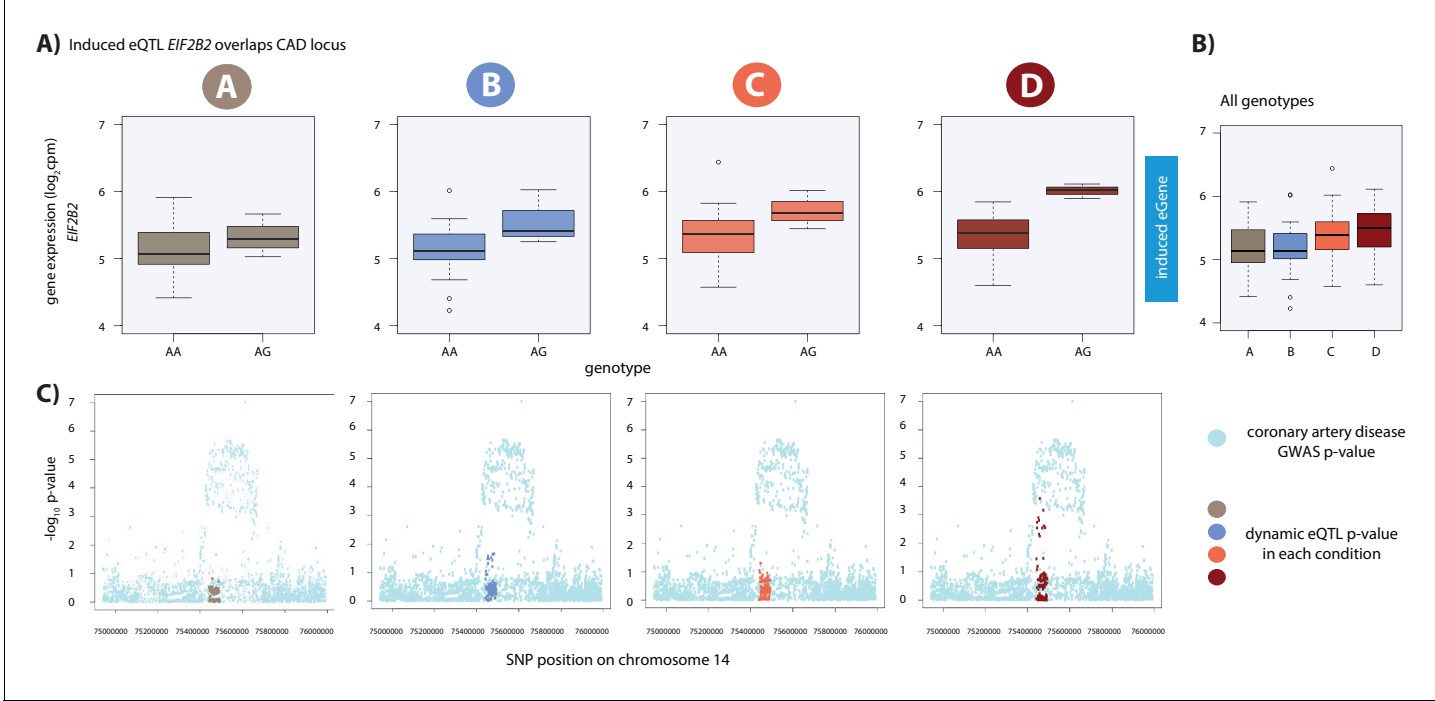

**Figure 7.** Dynamic eQTL for *EIF2B2* overlaps a coronary artery disease GWAS locus. (**A**) *EIF2B2* is a dynamic eGene. (**B**) Expression levels of *EIF2B2* during the course of the experiment following aggregation of all individuals. (**C**) Locus zoom plots in each condition illustrating the tested SNPs in the CARDIoGRAMplusC4D GWAS study (light blue), and the SNPs tested in our eQTL analyses in each condition (A: brown, B: blue, C: coral, D: red). All tested SNPs within a 1 Mb region around the dynamic eQTL SNP on chromosome 14 are shown.

The online version of this article includes the following figure supplement(s) for figure 7:

**Figure supplement 1.** Dynamic eQTLs associate with SNPs and genes implicated in complex disease.
**Figure supplement 2.** *EIF2B2* eSNP is in high LD with the lead GWAS variant associated with CAD at this locus.
**Figure supplement 3.** Hypoxic eQTL *CARM1* associates with MI.
**Figure supplement 4.** Power to detect eQTLs and false positive rate to call dynamic eQTLs.

of identified eQTL SNPs correspond to SNPs associated with cardiac traits, thus linking specific genes to organismal-level phenotypes. A compelling example is the association between *MYOZ1* expression and atrial fibrillation (*Lin et al., 2014*; *Sigurdsson et al., 2017*). Across tissues, the GTEx consortium reported that ~ 50% of eQTLs are also associated with variation in other measured complex traits (*GTEx Consortium et al., 2017*), and Heinig et al. have shown that 20% of left ventricle eQTLs relate to heart-associated loci (*Heinig et al., 2017*). However, these variants, identified in healthy individuals, are unlikely to represent all genetic variants that have consequences on disease. Indeed, Heinig et al. identified 100 dilated cardiomyopathy–specific eQTLs (not seen in healthy individuals) in a case-control study of 97 individuals with dilated cardiomyopathy and 108 healthy donors (*Heinig et al., 2017*). Similarly, by collecting samples pre- and post-surgically-induced ischemia, Stone et al. identified genetic associations that are only detected under stress (*Stone et al., 2019*). Although these studies provide a set of gene targets for further investigation, there are many loci that remain unexplained.

The heart is a complex tissue consisting of multiple cell types. The effects of some genetic variants in specific cell types might well be masked when considering heterogeneous tissue samples. As we are now able to direct iPSCs toward a cardiac fate, we can test for genetic effects on specific cell types such as cardiomyocytes (*Panopoulos et al., 2017*). As one would expect, iPSC-CMs are better suited to study cardiovascular traits than the immortalized B-cells or iPSCs from which they are derived (*Banovich et al., 2018*). However, given the high degree of eQTL sharing across diverse tissues (GTEx), identifying eQTLs in the disease-relevant terminal cell type at steady state may not give

substantial insight into disease biology. A significant advantage of using iPSC-CMs is that these cells provide a system to interrogate gene expression dynamics.

Cellular stressors that perturb gene expression levels and the cell state can unmask additional layers of regulatory variation (*Alasoo et al., 2018*; *Barreiro et al., 2012*; *Çalışkan et al., 2015*; *Kim-Hellmuth et al., 2017*; *Knowles et al., 2018*; *Manry et al., 2017*; *Nédélec et al., 2016*). Intermediate developmental cell states can similarly provide insight into GWAS loci where eQTL analysis in terminal cell types cannot (*Strober et al., 2019*). Further evidence for the notion that steady state eQTLs may have limited applicability to disease states comes from our previous work, where we used a comparative evolutionary approach to investigate the response to stress. We showed that genes that respond to oxygen stress in iPSC-CMs from both humans and chimpanzees, and are therefore likely relevant for disease, are depleted for eQTLs identified in heart tissue compared to genes that do not respond in either species (*Ward and Gilad, 2019*).

In the current study, by subjecting iPSC-CMs from a panel of individuals to perturbation (oxygen deprivation), we were able to identify a handful of potentially interesting trait-relevant loci. Using a broad GWAS catalog, we found one of our dynamic eQTL SNPs (rs8053350) to be associated with varicose veins, and the level of *RNF166* expression (*Fukaya et al., 2018*). This SNP falls within an intron of the *PIEZO1* gene. Varicose veins are associated with a risk for developing deep vein thrombosis and other vascular diseases (*Chang et al., 2018*). When we performed an analogous analysis focused on genes previously associated with three relevant traits – MI, heart failure and stroke, we identified a novel heart eGene, *ZC3HC1,* encoding the NIPA protein, which is implicated in MI, coronary artery disease, and ischemic stroke (*IBC 50K CAD Consortium, 2011*; *Nikpay et al., 2015*; *Schunkert et al., 2011*). This dynamic eQTL SNP is also associated with bronchodilator responsiveness in chronic obstructive pulmonary disease (*Hardin et al., 2016*). Using full summary statistics from a CAD and MI GWAS (*Stitziel et al., 2016*; *Nikpay et al., 2015*), we found minimal overlap between significant GWAS loci and significant or dynamic eSNPs. This may be due in part to differences in genetic ancestry and LD structure between the predominant GWAS population (of European descent) and our eQTL study population (Yoruba). Nevertheless, we highlight two interesting loci *EIF2B2* and *CARM1*. While the *EIF2B2* locus does not reach genome-wide significance in the GWAS analysis, integration with our dynamic eQTL data highlights this region as being potentially relevant to the disease. Indeed, *EIF2B2*, a GTP exchange protein, has been predicted to be a gene target in CAD using summary data-based Mendelian randomization (*Pavlides et al., 2016*). *CARM1,* also known as *PRMT4,* has previously been implicated in myocardial infarction and apoptosis in mice (*Wang et al., 2019*).

## Mechanisms behind response genes and dynamic eQTLs

Changes in gene expression can associate with other molecular-level phenotypes. The response to hypoxia is mediated by the HIF1α transcription factor (*Samanta and Semenza, 2017*), but given that there are hundreds of HIF1α binding locations and thousands of differentially expressed genes, regulation by this factor alone cannot directly explain all the transcriptional changes. We explored two additional molecular phenotypes in the context of oxygen deprivation – the locations and level of accessibility of open chromatin regions, and DNA methylation levels. We did not find either to contribute substantially to the gene expression response we observed. There are minimal changes in accessibility following hypoxia, which is in contrast to observations of studies that considered stimulation of immune cell types (*Alasoo et al., 2018*; *Calderon et al., 2019*; *Pacis et al., 2015*). This could reflect cell type specificity in response to stress, or the specificity of the cellular response to different stressors. Indeed, despite large gene expression changes in response to various stimulants in endothelial cells, there are a relatively small number of differentially accessible regions (*Findley et al., 2019*). We speculate that the transcriptional response to oxygen stress could result in the induction of transcription factors, which bind already accessible regions of open chromatin, and that cells are primed for a quick response to this universal cellular stress. Indeed, it has been shown that chromatin contacts exist between HIF1α binding sites and hypoxia-inducible genes in the normoxic state (*Platt et al., 2016*). Conversely, it has been suggested that hypoxia results in the induction of HIF1α, and significant changes in histone methylation (*Batie et al., 2019*). As we did not measure histone marks in our system, these changes may occur in the absence of chromatin accessibility changes, but we also cannot rule out the possibility that the choice of a single timepoint

following 6 hr of hypoxia, or insufficient statistical power in our sample size, contributed to the minimal differences in accessibility that we observed.

Using an approach designed to measure small effect sizes between conditions, we did identify a set of 831 DARs between hypoxia and short-term re-oxygenation that are enriched for marks of active chromatin, CpG islands, and TEs. These regions do not appear to explain many of the gene expression differences we observed. Hypoxia and oxidative damage are likely to also affect the genome in ways that do not directly impact gene expression. Indeed, the distribution of oxidative DNA damage sites varies across the genome following stress such that TEs and active chromatin regions are enriched for DNA damage, while promoters are depleted (*Poetsch et al., 2018*). We found enrichment for TEs, specifically Alu SINE elements, in DARs. Interestingly, TEs, and DNA transposons in particular, are also enriched in regions that become accessible in macrophages in response to bacterial infection; suggesting sequence-specific effects of TEs in response to different cellular stressors (*Bogdan et al., 2020*). Alu elements have previously been found to associate with the response to stress in other contexts. Serum starvation induces binding of TFIIIC, which recruits RNA polymerase III, to Alu elements (*Ferrari et al., 2020*), and heat shock increases chromatin accessibility around Alu elements (*Kim et al., 2001*).

There are several studies, which suggest that DNA methylation levels are dynamic and change in response to stressors such as hypoxia. We did not find any notable differences in DNA methylation levels pre- and post-hypoxia and re-oxygenation, which suggests that like chromatin accessibility, DNA methylation levels do not make large contributions to changes in gene expression levels or the appearance of dynamic eQTLs in our system. Many of the DNA methylation changes that have been described in response to hypoxia occur in chronic and intermittent hypoxia, and not acute hypoxia as investigated in our study (*Hartley et al., 2013*; *Robinson et al., 2012*; *Watson et al., 2014*). DNA methylation levels are also altered in response to other stressors such as bacterial infection (*Pacis et al., 2015*); however, the importance of timing is highlighted by the fact that, in this system, gene expression responses precede DNA methylation changes (*Pacis et al., 2019*). It is also important to note that our study considers baseline oxygen levels to be 10% oxygen, which is closer to physiological oxygen levels (5–13%) than atmospheric oxygen levels (21%; *Brahimi-Horn and Pouys-ségur, 2007*; *Carreau et al., 2011*; *Jagannathan et al., 2016*). Most studies define normoxia as 21% oxygen saturation, and while this likely leads to larger effect size differences in known hypoxia response genes following hypoxia, these comparisons may not give meaningful insight into the in vivo state.

One can speculate about different mechanisms that might lead to the appearance or disappearance of dynamic eQTLs. In the context of the immune response, it has been shown that the same response variants affect both gene expression and chromatin accessibility (*Alasoo et al., 2018*). This is in line with the general notion that changing cellular environments results in differences in chromatin accessibility at transcription factor binding sites, which leads to gene expression changes. We found that this does not appear to be a major mechanism in our system as there are minimal changes in accessibility following hypoxia. We observed that there is an enrichment of response genes amongst dynamic eQTLs suggesting that the change in environment results in a change in expression levels that is dependent on the associated genotype. We also find enrichment for TFs amongst response genes and dynamic eQTLs, suggesting that dynamic eQTLs can appear through secondary *trans* effects.

## Potential limitations of our model

To understand the effects of genetic variation on human heart tissue, and how this variation might contribute to the MI and I/R injury etiologies of CVD, we carefully perturbed oxygen levels that cardiomyocytes in culture are exposed to. This in vitro approach is by design a model system, and therefore will likely not fully recapitulate the in vivo state. However, we previously found that out of 2,549 genes that respond to hypoxia in iPSC-CMs from humans and chimpanzees, only 16% are differentially expressed between iPSC-CMs and heart tissue (*Pavlovic et al., 2018*; *Ward and Gilad, 2019*). This suggests that our in vitro system is applicable to heart tissue. There is still a possibility that the dynamic eQTLs that we identify in our in vitro system are not physiologically relevant.

Our study comprised a small number of individuals (15), far fewer than what is typical for identifying eQTLs. Our work is therefore a first step toward understanding the effects of genetic variation on gene expression in response to stress. Nevertheless, with a small number of individuals we were

able to identify a couple of hundred dynamic eQTLs that are revealed or suppressed under stress, suggesting that this paradigm is worth exploring further in larger cohorts. Under the simplifying assumption of a single causal variant, we determined that we have ~6% power to detect an effect which explains ~38% of the heritability, and an equal false positive rate to call it a dynamic eQTL (see Materials and methods; *Figure 7—figure supplement 4*). The median eQTL heritability over all genes is 0.16, as previously reported by *Gusev et al., 2016*; *Wheeler et al., 2016*, which illustrates the relative power of this study compared to the GTEx and Depression Genes and Networks studies (*Battle et al., 2014*) for eQTL mapping. This suggests that the impact of stress on genotype-dependent effects on gene expression will be even greater in studies which have higher power to detect smaller effects of genotype. For perspective, early eQTL studies were similarly powered to our study, using 70 individuals; yet these studies still led to important insights opening an avenue of research focused on assaying the consequences of genetic variation by RNA-seq (*Pickrell et al., 2010*).

In summary, there have been few studies assessing the effects of genetic variation in response to CVD-relevant perturbations in cardiomyocytes. Here, we profiled the response to oxygen deprivation in cardiomyocytes from a panel of genotyped individuals. We find that eQTLs can appear and disappear in response to oxygen deprivation, and that some of these eQTLs have effects on relevant complex traits and disease.

# Materials and methods

## Key resources table

| Reagent type (species) or resource | Designation | Source or reference | Identifiers | Additional information |
|---|---|---|---|---|
| Cell line (*H. sapiens*, Female) | 18499 iPSC | 10.1101/gr.224436.117 | | Derived from HapMap Yoruba LCL |
| Cell line (*H. sapiens*, Female) | 18505 iPSC | 10.1101/gr.224436.117 | | Derived from HapMap Yoruba LCL |
| Cell line (*H. sapiens*, Female) | 18511 iPSC | 10.1101/gr.224436.117 | | Derived from HapMap Yoruba LCL |
| Cell line (*H. sapiens*, Female) | 18520 iPSC | 10.1101/gr.224436.117 | | Derived from HapMap Yoruba LCL |
| Cell line (*H. sapiens*, Female) | 18852 iPSC | 10.1101/gr.224436.117 | | Derived from HapMap Yoruba LCL |
| Cell line (*H. sapiens*, Female) | 18855 iPSC | 10.1101/gr.224436.117 | | Derived from HapMap Yoruba LCL |
| Cell line (*H. sapiens*, Female) | 18858 iPSC | 10.1101/gr.224436.117 | | Derived from HapMap Yoruba LCL |
| Cell line (*H. sapiens*, Female) | 18870 iPSC | 10.1101/gr.224436.117 | | Derived from HapMap Yoruba LCL |
| Cell line (*H. sapiens*, Female) | 18912 iPSC | 10.1101/gr.224436.117 | | Derived from HapMap Yoruba LCL |
| Cell line (*H. sapiens*, Male) | 19098 iPSC | 10.1101/gr.224436.117 | | Derived from HapMap Yoruba LCL |
| Cell line (*H. sapiens*, Male) | 19101 iPSC | 10.1101/gr.224436.117 | | Derived from HapMap Yoruba LCL |

*Continued on next page*

*Continued*

| Reagent type (species) or resource | Designation | Source or reference | Identifiers | Additional information |
|---|---|---|---|---|
| Cell line (*H. sapiens*, Female) | 19108 iPSC | 10.1101/gr.224436.117 | | Derived from HapMap Yoruba LCL |
| Cell line (*H. sapiens*, Female) | 19116 iPSC | 10.1101/gr.224436.117 | | Derived from HapMap Yoruba LCL |
| Cell line (*H. sapiens*, Female) | 19128 iPSC | 10.1101/gr.224436.117 | | Derived from HapMap Yoruba LCL |
| Cell line (*H. sapiens*, Female) | 19160 iPSC | 10.1101/gr.224436.117 | | Derived from HapMap Yoruba LCL |

## Samples

We randomly selected individuals from the Yoruba YRI HapMap population. iPSCs were reprogrammed from lymphoblastoid cell lines from these individuals (*Banovich et al., 2018*). We used fifteen biological replicates (individuals), and three technical replicates (independent cardiomyocyte differentiation and oxygen stress experiments) from three individuals. This number of individuals has been shown to be sufficient to be able to identify eQTLs in small sample sizes (*van de Geijn et al., 2015*). Experiments were designed and performed such that technical variables (such as sample processing batch) were not confounded with the variable of interest (condition). All cell lines tested negative for mycoplasma contamination.

## Cardiomyocyte differentiation from iPSCs

iPSCs were maintained in a feeder-independent state in Essential 8 Medium (A1517001, Thermo-Fisher Scientific, Waltham, MA, USA) with Penicillin/Streptomycin (30002, Corning, NY, USA) on Matrigel hESC-qualified Matrix (354277, Corning, Bedford, MA, USA) at a 1:100 dilution. Cells were passaged at ~70% confluence every 3–4 days with dissociation reagent (0.5 mM EDTA, 300 mm NaCl in PBS), and seeded with ROCK inhibitor Y-27632 (ab12019, Abcam, Cambridge, MA, USA).

Cardiomyocyte differentiations were performed largely as previously described (*Ward and Gilad, 2019*), except the duration and concentration of the Wnt agonist and antagonist differed for this panel of individuals, which included only human samples. Briefly, on Day 0, iPSC lines at 70–100% confluence in 100 mm plates were treated with 12 µM GSK3 inhibitor CHIR99021 trihydrochloride (4953, Tocris Bioscience, Bristol, UK) in 12 ml Cardiomyocyte Differentiation Media [500 mL RPMI1640 (15–040 CM ThermoFisher Scientific), 10 mL B-27 Minus Insulin (A1895601, ThermoFisher Scientific), 5 mL Glutamax (35050–061, ThermoFisher Scientific), and (5 mL Penicillin/Streptomycin)], and a 1:100 dilution of Matrigel. 24 hr later, on Day 1, the media was replaced with Cardiomyocyte Differentiation Media. 48 hr later, on Day 3, 2 µM of the Wnt inhibitor Wnt-C59 (5148, Tocris Bioscience), diluted in Cardiomyocyte Differentiation Media, was added to the cultures. Cardiomyocyte Differentiation Media was replaced on Days 5, 7, 10, and 12. Cardiomyocytes were purified by metabolic purification by the addition of glucose-free, lactate-containing media (Purification Media) [500 mL RPMI without glucose (11879, ThermoFisher Scientific), 106.5 mg L-Ascorbic acid 2-phosphate sesquimagenesium salt (sc228390, Santa Cruz Biotechnology, Santa Cruz, CA, USA), 3.33 ml 75 mg/ml Human Recombinant Albumin (A0237, Sigma-Aldrich, St Louis, MO, USA), 2.5 mL 1 M lactate in 1 M HEPES (L(+)Lactic acid sodium (L7022, Sigma-Aldrich)), and 5 ml Penicillin/Streptomycin] on Days 14, 16, and 18. A total of 1.5 million cardiomyocytes were re-plated per well of a six-well plate on Day 20 in Cardiomyocyte Maintenance Media (500 mL DMEM without glucose [A14430-01, Thermo-Fisher Scientific], 50 mL FBS [S1200-500, Genemate], 990 mg Galactose [G5388, Sigma-Aldrich], 5 mL 100 mM sodium pyruvate [11360–070, ThermoFisher Scientific], 2.5 mL 1 M HEPES [SH3023701, ThermoFisher Scientific], 5 mL Glutamax [35050–061, ThermoFisher Scientific], 5 mL Penicillin/Streptomycin). iPSC-CMs were matured in culture for a further 10 days with Cardiomyocyte Maintenance Media replaced on Days 23, 25, 27, 28, and 30.

On Day 25, iPSC-CMs were transferred to a 10% oxygen environment (representative of in vivo levels) in an oxygen-controlled incubator (HERAcell 150i $CO_2$ incubator, ThermoFisher Scientific). From Day 27 onwards, iPSC-CMs were pulsed at a voltage of 6.6 V/cm, frequency of 1 Hz, and pulse frequency of 2 ms using an IonOptix C-Dish and C-Pace EP Culture Pacer to further mature the cells and synchronize beating.

## Flow cytometry

Purity of the cardiomyocyte cultures was assessed ~Day 30 as previously described (*Ward and Gilad, 2019*). Briefly, cells were stained with Zombie Violet Fixable Viability Kit (423113, BioLegend), and PE Mouse Anti-Cardiac Troponin T antibody (564767, clone 13–11, BD Biosciences, San Jose, CA, USA), and analyzed on a BD LSRFortessa Cell Analyzer together with negative control samples of iPSCs, and iPSC-CMs that are incubated without the troponin antibody, or without either the troponin antibody or viability stain.

## Hypoxia experiment

On Day 31/32, iPSC-CMs were subjected to the hypoxia experiment. At time = 0, condition A samples remained at 10% $O_2$ (normoxia), while samples for conditions B, C and D were transferred to an incubator set at 1% $O_2$ (hypoxia). After 6 hr, conditions A and B were harvested, while plates C and D were returned to normoxic oxygen conditions. Plate C was harvested 6 hr following the hypoxic treatment, and Plate D was harvested 24 hr following the hypoxic treatment. Oxygen levels, experienced by the cells in culture, were measured in cultures from each experimental batch using an oxygen-sensitive sensor (SP-PSt3-NAU-D5-YOP, PreSens Precision Sensing GmbH, Regensburg, Germany), optical fiber (NWDV29, Coy, Grass Lake, MI, USA), and oxygen meter (Fibox 3 Transmitter NWDV16, Coy).

## Material collection

### Cell culture media for ELISA and cytotoxicity assays

Aliquots of cell culture media from each experiment were centrifuged at 10,000 rpm for 10 min at 4°C to remove cellular debris. The supernatant was stored at −80°C until further use.

### Nuclei for ATAC-seq

Cardiomyocytes from each well of a six-well plate were washed twice with cold PBS on ice before collection by manual scraping in 1.5 ml PBS. A total of 200 µl of cells were pelleted by centrifugation at 500 g for 5 min. Cell pellets were re-suspended in 50 µl cold ATAC-seq lysis buffer (10 mM Tris-HCl pH 7.4, 10 mM NaCl, 3 mM $MgCl_2$, 0.1% Igepal CA630, $dH_2O$). Nuclei were pelleted by centrifugation at 500 g for 5 min at 4°C. Nuclei were re-suspended in 50 µl transposition mix (25 µl 2xTD buffer, 2.5 µl Tn5 transposase, 22.5 µl nuclease-free $dH_2O$) from the Nextera DNA sample kit (FC-121–1031, Illumina). The transposition reaction was performed at 37°C for 30 min. Transposed DNA was purified with Qiagen MinElute Kit (28004, Qiagen, MD, USA), re-suspended in 12 µl elution buffer, and stored at −20°C.

### Cell pellets for RNA-seq and DNA methylation arrays

Cells from each well of a six-well plate were washed twice with cold PBS on ice before collection by manual scraping in 1.5 ml PBS. Cells were pelleted by centrifugation at 7,000 rpm for 8 min at 4°C, flash-frozen and stored at −80°C.

## RNA/DNA extraction

RNA and DNA were extracted from the same frozen cell pellets using the ZR-Duet DNA/RNA Mini-Prep kit (D7001, Zymo, CA, USA) according to the manufacturer's instructions. All four conditions from three or four individuals were extracted in the same batch. RNA samples had a median RIN score of 8.5 with similar median scores across conditions (A = 9.4, B = 8.7, C = 9.1, D = 9.2) (*Supplementary file 1*, *Figure 2—figure supplement 1*). All samples had RIN scores greater than eight except for two samples: 18852A RIN = 6, 18852D RIN = not determined. 18852A is an outlier sample in *Figure 2—figure supplement 2*. Given that both of these samples come from an

individual for which we have replicate samples, these particular samples were not selected for the differential expression analysis and eQTL analysis that are only able to handle one biological replicate.

## RNA-seq library preparation

We prepared RNA-seq libraries from 15 individuals, and three replicates from three individuals (18852, 18855, 18511). A toal of 500 ng of RNA were used to prepare sequencing libraries using the Illumina TruSeq RNA Sample Preparation Kit v2 (RS-122–2001 and −2002, Illumina). Libraries were pooled into five master mixes containing 12 or 16 samples. Each pool was sequenced 50 bp, single-end on the HiSeq2500 or HiSeq4000 according to the manufacturer's instructions.

## DNA methylation array

We measured DNA methylation in 13 of the 15 individuals we collected gene expression data for (we do not have data from any condition for individuals 18870 and 19116; *Supplementary file 1*). As we did for the RNA-seq data, we collected three replicates for three individuals; except for individual 18852 which only had two replicates. Nine chips (eight samples per chip) with 60–1000 ng DNA were bisulfite-converted and processed on an Illumina Infinium MethylationEPIC array at the University of Chicago Functional Genomics facility.

## ATAC-seq

We performed ATAC-seq in 14 of the 15 individuals we had gene expression data for (we do not have data from any condition for individual 18858; *Supplementary file 1*). One sample was collected for each individual and condition. ATAC-seq libraries were prepared using the Illumina Nextera DNA sample kit. Libraries were amplified for 10–16 cycles depending on the amplification rate of each library. Each library was amplified in a PCR reaction containing 10 µl DNA, 10 µl dH$_2$O, 15 µl NMP (PCR master mix), 5 µl PPC (PCR primer cocktail), 5 µl index N5, and 5 µl index N7. PCR conditions were set at 72°C for 5 min, 98°C for 30 s, 98°C for 10 s, 63°C for 30 s, 72°C for 1 min, repeat steps 3–5 4x and hold at 4°C. The number of cycles per library was determined using a qPCR side reaction as described in *Buenrostro et al., 2013*. Libraries were purified using Agencourt AMPure XP beads (A63880, Beckman Coulter, IN, USA), and bioanalyzed to determine library quality. Twelve or 16 samples were pooled together to generate four master mixes. Each master mix was sequenced 50 bp paired-end on the HiSeq4000 according to the manufacturer's instructions.

## Lactate dehydrogenase activity assay

Lactate dehydrogenase activity (LDH) was measured in 5 µl cell culture media using the Lactate Dehydrogenase Activity Assay Kit (MAK066, MilliporeSigma, MO, USA) according to the manufacturer's instructions. Each sample was assayed in triplicate. LDH activity was measured as the difference in absorbance prior to the addition of the substrate, and 10 min after the initiation of the enzymatic reaction, calculated relative to a standard curve. Measurements are standardized relative to A, and reported as A (A-A), B (B-A), C (C-B) and D (D-B).

## BNP ELISA

A total of 125 µl of cell culture media was assayed to quantify the level of secreted BNP using the Brain Natriuretic Peptide EIA kit (RAB0386, MilliporeSigma). Each sample was assayed in duplicate on two 96-well plates. BNP levels were quantified relative to a standard curve using 4- and 5-parameter logistic models using the R package drc. Measurements are standardized relative to A, and reported as A (A-A), B (B-A), C (C-B), and D (D-B).

## RNA-seq analysis

Reads were aligned to hg19 using subread align (*Liao et al., 2013*). The mapped reads were then reprocessed to reduce reference bias for downstream analyses using the WASP pipeline (*van de Geijn et al., 2015*). Briefly, reads overlapping polymorphisms segregating in our population were remapped to the genome using the true read, and a version of the read with the alternative allele. Only reads that mapped uniquely to the same locations with both possible alleles were kept. The median number of reads across conditions was similar (A: 34,353,716; B: 33,493,298; C: 33,883,532;

D: 38,147,083). The number of filtered reads mapping to genes was quantified using featureCounts within subread (*Liao et al., 2014*). We obtained measurements for 19,081 genes. Sample 18852A was an outlier when considering read count correlations between pairs of samples, and was therefore removed prior to subsequent analyses.

## Differential expression analysis

We selected autosomal genes for downstream analysis (18,226). $Log_2$-transformed counts per million were calculated (*Robinson et al., 2010*), and genes with a mean $log_2$cpm < 0 were excluded. We used the fact that we have replicate data from three individuals to remove unwanted variation in our data. We used the RUVs function in the RUVSeq package in R (*Risso et al., 2014*) to identify such factors. By manual inspection, our data segregated by individual or condition after correction with four factors. For the differential expression analysis, we excluded sample replicate one to avoid the outlier sample and randomly selected replicate two, instead of replicate three, for individuals with replicate samples. We used the RUV factors as covariates in our differential expression analysis using the TMM-voom-limma pipeline (*Law et al., 2014*; *Robinson et al., 2010*; *Smyth, 2004*). We used fixed effects for each condition (A, B, C, D), the RUVs factors as covariates, and a random effect for individual, which was implemented using duplicateCorrelation. Genes with a Benjamini and Hochberg FDR < 0.1 are classified as differentially expressed (*Benjamini and Hochberg, 1995*).

## Gene expression trajectory analysis

To identify response genes, we used the Cormotif package in R (*Wei et al., 2015*) to jointly model pairs of tests. We used TMM-normalized $log_2$cpm values as input and considered the following pairs of tests: A vs B, B vs. C, and B vs. D to determine which genes are changing their expression during the course of the experiment. The best fit was determined to correspond to two correlation motifs or clusters using BIC and AIC. We classified genes as response genes if the probability of differential expression between conditions was >0.5 in all pairs of tests.

## eQTL identification

To map eQTLs, we analyzed the same samples considered in the differential expression analysis. Given the sample size in this study, we utilized the combined haplotype test (CHT) to identify eQTLs (*van de Geijn et al., 2015*). This test models both allelic imbalance and total read depth at a region to identify QTLs. We require 50 total counts and 10 allele-specific counts for each gene, and tested variants 25 kb upstream and 25 kb downstream of the TSS, resulting in 1,040,874 shared tests (A: 1,215,476; B: 1,211,099, C: 1,224,612, D: 1,201,078). As previously reported, we found that null p-values for the CHT were not calibrated in our data. To calibrate the p-values, we estimated the null distribution of the CHT by permuting the data 100 times and fitting a Beta distribution to the permuted p-values for each SNP-gene pair (previously proposed by *Delaneau et al., 2017*; *Ongen et al., 2016*). We then computed an adjusted p-value for each SNP-gene pair by taking the CDF of the fitted Beta distribution, evaluated at the reported CHT p-value.

To call significant eQTLs, we estimated q-values for the set of adjusted p-values for each phenotype, and took tests with q < 0.1. The number of eGenes in each condition was determined by taking the most significant SNP-gene association in each condition (i.e. the top SNP). We defined dynamic eQTLs as either: (1) significant only in A (q<0.1 in A and permutation-adjusted p>0.15 in B and C and D; suppressed eQTL); (2) significant in at least one of B, C, or D (q < 0.1) and not nominally significant in A (adjusted p>0.15; induced eQTL).

## Power analysis

For QTL mapping, we assume a linear model

$$
\begin{aligned}
y_i &= x_i\beta + \epsilon_i \\
\epsilon_i &\sim \mathcal{N}(0, \sigma^2)
\end{aligned}
$$

where $y_i$ denotes the phenotype of individual $i$ and $x_i$ denotes the genotype of individual $i$ at a single SNP of interest. We estimate an effect size $\beta$

$$\hat{\beta} \sim \mathcal{N}\left(\beta, \frac{\sigma^2}{n}\right)$$

where $n$ is the sample size. Let $\lambda = \beta/\sigma$ be the standardized effect size. Then,

$$\hat{\lambda} \sim \mathcal{N}\left(\lambda, \frac{1}{n}\right)$$

and

$$\mathrm{Power}(\lambda, \alpha, n) = \Phi\left(\Phi^{-1}(\alpha/2) + \lambda\sqrt{n}\right)$$

where $\alpha$ denotes the significance level and $\Phi$ denotes the standard Gaussian CDF. To simplify the analysis, we consider $\alpha = 0.05/20000 = 2.5 \times 10^{-6}$ (i.e. Bonferroni correction; this is equivalent to controlling the FDR when all tests are null, and is conservative otherwise). Assume there is a single causal variant. Then, the phenotypic variance explained is:

$$h^2 = \frac{\lambda^2}{\lambda^2 + 1}$$

We defined a dynamic eQTL as either significant only in A, or significant (after Bonferroni correction, in this analysis) in one of B, C, or D and not significant in A. To estimate the false-positive rate of dynamic eQTL calling, we asked what was the probability of a SNP passing this definition, assuming the standardized effect size $\lambda$ was *identical* in all four conditions. We then computed phenotypic variance explained, power to detect an eQTL, and false-positive rate to call a dynamic eQTL for every choice of standardized effect size $\lambda$.

## Overlapping response genes and eGenes with existing gene sets

### Gene ontology analysis

Gene set enrichment analysis was performed on response genes, and a background set of all expressed genes using the DAVID genomic annotation tool (*Huang et al., 2009a*; *Huang et al., 2009b*). GO Terms related to Biological Processes were selected, and those with a Benjamini-Hochberg controlled FDR < 0.05 were designated as significantly enriched. Each of the five significantly enriched processes relates to transcription ('DNA-templated transcription', 'DNA-templated regulation of transcription', 'DNA-templated negative regulation of transcription', 'negative regulation of transcription from RNA polymerase II promoter', 'positive regulation of transcription from RNA polymerase II promoter'). The most significantly enriched GO terms related to Molecular Functions include 'transcription factor activity, sequence-specific DNA binding', 'nucleic acid binding' and 'DNA binding'.

### Transcription factors

A list of 1,637 annotated human TFs was obtained from *Lambert et al., 2018*, and intersected with our gene sets.

### GTEx eQTLs

Three hundred and sixty-seven dynamic eSNPs were interrogated for overlap with eQTL data from left ventricle heart tissue and 49 other tissues assayed in GTEx v8 (http://www.gtexportal.org). A total of 326 dynamic eSNPs were tested in GTEx. We determined whether each dynamic eSNP was identified as a significant eQTL in left ventricle heart tissue or any other tissue. To define shared eSNPs we identified the 61 eGenes present in all four conditions. We then identified the most significant eSNP for that gene in condition A and determined whether this SNP was significantly associated with the expression of that gene in conditions B, C, and D. This yielded 20 shared eSNPs, 19 of which were tested by the GTEx consortium. We compared our dynamic eQTL and shared eQTL effect sizes to the effect size in GTEx left ventricle heart tissue. In this analysis, for the dynamic eQTLs we selected the condition with the largest effect size, and for the shared eQTLs we used the effect size in condition A to compare to the GTEx effect size.

## ATAC-seq analysis

Paired-end sequencing reads were aligned to hg19 using bowtie2 with default settings (*Langmead and Salzberg, 2012*). Reads were filtered using Picard Tools (https://broadinstitute.github.io/picard/) to remove duplicate reads, and reads mapping to the mitochondrial genome. Reads were then remapped using the WASP pipeline as described above. We retained a similar median number of reads across conditions (A: 28,998,060; B: 33,662,261; C: 30,161,640; D: 34,534,416). Across conditions, there is no significant difference in the number of mapped reads, number of regions identified, or fraction of reads mapped to open chromatin regions (*Figure 4—figure supplement 2A–C*). All libraries, across conditions, show the expected fragment size distribution, enrichment of reads at transcription start sites (TSS), and footprints at well-defined CTCF motifs (*Figure 4—figure supplement 2D–F*). Correlation analysis of read counts between pairs of samples revealed clustering by individual and condition (*Figure 4—figure supplement 3*). As expected, the correlation of read counts between samples at the 10,633 regions overlapping the TSS is higher than the correlation across all regions (median rho = 0.83 vs. 0.56). Pairs of samples from the same condition are marginally more correlated in their accessibility profiles than pairs of samples across all conditions (median rho = 0.84 vs. 0.83 at the TSS).

## Identification of accessible chromatin regions

To generate a unified list of regions with accessible chromatin across conditions and samples, we first used MACS2 (*Zhang et al., 2008*) to identify peaks within each sample independently. Next, we used BEDtools (*Quinlan and Hall, 2010*) with the multiIntersectBed function to identify overlapping peaks within each condition separately. Within each condition, we retained peaks with support from more than three individuals and used the mergeBed function to create a condition-specific consensus. We then combined and merged the bed files across the four conditions to make a final consensus file containing all the filtered accessible regions. The number of reads mapping to accessible chromatin regions was quantified using featureCounts within subread (*Liao et al., 2014*).

## Identification of differentially accessible regions (DARs)

The 128,673 open chromatin regions associated with count data were filtered to include only those regions on the autosomes, and those which had mean $\log_2$cpm values > 0 for each region. First, to identify differentially accessible regions we used the same limma framework described above for the RNA-seq data. To test for differences between conditions, a linear model with a fixed effect for condition was used together with a random effect for individual. We did not identify any significantly differentially accessible regions with a Benjamini and Hochberg FDR < 0.1. To identify regions with small effect size differences between conditions, we used an adaptive shrinkage method implemented in the ashr package in R (*Stephens, 2017*). We used the regression estimates (regression coefficients, posterior standard errors, and posterior degrees of freedom) generated by limma to calculate a posterior mean (shrunken regression coefficients), FDR, and False Sign Rate (FSR, probability that the sign of the effect size is wrong). We considered regions to be differentially accessible at FSR <0.1. We denote regions that are not differentially accessible as constitutively accessible regions.

## Overlap of DARs with genomic features

### TSS

Transcription start sites were obtained from the UCSC Table Browser (http://genome.ucsc.edu/cgi-bin/hgTables) using 'txStart' from Ensembl genes (*Karolchik et al., 2004*). TSS were defined based on the TSS of the 5' most transcript on the sense strand and 3' most transcript on the anti-sense strand. TSS regions, and subsequent genomic features, were intersected with DARs and constitutively accessible regions requiring a 1 bp overlap using bedtools intersect (*Quinlan and Hall, 2010*).

### Histone marks

We obtained histone mark data (.bed files) for human heart tissue from the ENCODE consortium (*ENCODE Project Consortium, 2012*; *Davis et al., 2018*) ENCODE portal, (https://www.encode-project.org). We selected H3K4me3 (Experiment ENCSR181ATL), H3K4me1 (Experiment ENCSR449FRQ), H3K36me3 (Experiment ENCSR799KLF), H3K27me3 (Experiment: ENCSR613PPL),

and H3K9me3 (Experiment ENCSR803MVC) ChIP-seq data from heart left ventricle tissue from a 51-year-old female individual (Biosample ENCBS684IAD).

## Transcription factor binding locations

We obtained ChIP-seq data for the hypoxia-responsive factors HIF1α and HIF2α assayed in the MCF-7 breast cancer cell line (*Schödel et al., 2011*), and E2F4 in the GM06990 lymphoblastoid cell line (*Lee et al., 2011*). Genome co-ordinates of the 356 HIF1α-, 301 HIF2α- and 16,245 E2F4-bound regions were converted from hg18 to hg19 using the liftOver tool in the Galaxy platform (http://galaxyproject.org/; *Afgan et al., 2018*).

## Motif enrichment analysis in DARs

We obtained sequences for all accessible regions and differentially accessible regions using the Galaxy platform (*Afgan et al., 2018*). We used the MEME-ChIP tool within The MEME Suite (*Bailey et al., 2009*; *Machanick and Bailey, 2011*) in Differential Enrichment mode to identify motifs differentially enriched in DARs compared to all accessible regions using a hypergeometric distribution.

## TEs

We obtained repeat annotations from the RepeatMasker track (*Jurka, 2000*; *Smith et al., 2010*) from the UCSC Table browser (*Karolchik et al., 2004*). We intersected the Repeatmasker track with our accessible regions and reported those elements where 50% of their length overlaps a DAR or constitutively accessible region. We stratified TEs by TE class: LINE, SINE, DNA, and LTR, and then by TE family and type within the SINE class.

## CpG islands

We obtained CpG island annotations from the UCSC Table Browser, and overlapped these regions with DARs and constitutively accessible regions.

## caQTL identification

The caQTLs were identified in the same manner as described for the eQTLs. However, in the caQTL analyses, we limited tested SNPs to those falling within the 128,672 accessible regions, as opposed to testing variants within 25 kb of the region as we did for eQTLs. Dynamic caQTLs were identified as for dynamic eQTLs.

## DNA methylation analysis

To allow for accurate quantification of DNA methylation levels we removed probes overlapping SNPs with a minor allele frequency of >0.1, and only retained probes with a detection p-value of >0.75 across samples. Beta-values (ratio of methylated probe intensity and overall probe intensity, and bounded between 0 and 1) were quantile normalized using lumiN, and, when appropriate, converted to M-values ($\log_2$ ratio of intensities of methylated probe versus unmethylated probe) using lumi (*Du et al., 2008*).

The methylation level of CpGs coincides with the expected distribution based on their annotated genomic location that is low levels of DNA methylation in CpG islands, and higher levels in CpG island shores, and CpG island shelves respectively (*Figure 6—figure supplement 1*). Correlation analysis across all pairs of samples, including replicate samples, reveals clustering primarily by individual rather than condition (*Figure 6—figure supplement 2*).

To measure the DNA methylation level at gene set promoters, we selected CpGs 200 bp upstream of the TSS (TSS200 defined on the array). We considered all CpGs when overlapping with DARs.

## Identification of differentially methylated CpGs (DMCpGs)

Differentially methylated CpGs were identified using the same limma framework as described for the RNA-seq data. Analysis was run using both Beta-values and M-values.

## Variance partition of three molecular phenotypes

To identify the proportion of variance explained by individual, condition and replicate in the gene expression, chromatin accessibility, and DNA methylation data, we used a linear mixed model with a random effect for all of the variables. The variance was normalized to sum to one and the proportion of variance attributed to each variable was calculated at each locus. We used a t-test to compare the mean proportion of variance explained by the same variable between data types.

## Integration with GWAS-implicated variants and genes

We intersected the Reference SNP cluster ID of our dynamic QTLs with the 158,654 SNPs in the NHGRI-EBI GWAS Catalog available from the UCSC Table Browser (*Buniello et al., 2019*) in August 2019.

We also considered the 'mapped genes' results from GWAS from thee relevant traits: myocardial infarction (EFO_0000612, 89 genes), heart failure (EFO_0003144, 164 genes) and stroke (EFO_0000712, 255 genes), downloaded from the NHGRI-EBI GWAS Catalog in August 2019. Gene lists were intersected with our response eGenes.

Full GWAS summary statistics for CAD and MI were obtained from the CARDIoGRAMplusC4D Consortium (*Stitziel et al., 2016*; *Nikpay et al., 2015*). We tested if the lead eSNP of the dynamic eQTLs, and all significant eQTLs in each condition is associated with CAD or MI. We created locus zoom plots in each condition by plotting the -$\log_{10}$ p-values of the GWAS and eQTL SNPs in a 1 Mb window around eSNPs. We calculated LD from the YRI population using the LDlink program (*Machiela and Chanock, 2015*). We used the LDmatrix function to generate the LD matrix plots choosing all tested SNPs 5 kb upstream and downstream from the eSNP or lead GWAS variant. To generate the $R^2$ value and p value of the LD between the eSNP and GWAS variant we used the LDpair function.

## Data access

All RNA-seq, ATAC-seq and DNA methylation data have been deposited in the Gene Expression Omnibus (https://www.ncbi.nlm.nih.gov/geo/) under accession number GSE144426.

## Acknowledgements

We thank Kristen Patterson and Amy Mitrano for experimental assistance, David Knowles and Benjamin Strober for preliminary data exploration, all members of the Gilad lab and Luis Barreiro for helpful discussions, and Natalia Gonzales for comments on the manuscript. We thank the Genomics Core Facility at the University of Chicago for sequencing the libraries and processing the DNA methylation arrays. We thank The Genotype-Tissue Expression (GTEx) Project, supported by the Common Fund of the Office of the Director of the National Institutes of Health, and by NCI, NHGRI, NHLBI, NIDA, NIMH and NINDS, for providing data. The data used for the analyses described in this manuscript were obtained from the GTEx portal v8 on December 30th 2020. We thank the ENCODE Consortium and the Bernstein Lab at Broad for generating and making the histone mark ChIP-seq data available.

## Additional information

### Funding

| Funder | Grant reference number | Author |
|---|---|---|
| National Heart, Lung, and Blood Institute | HL092206 | Yoav Gilad |
| EMBO | Long-Term Fellowship ALTF 751-2014 | Michelle C Ward |
| National Institute on Aging | F31 AG044948 | Nicholas E Banovich |

The funders had no role in study design, data collection and interpretation, or the decision to submit the work for publication.

## Author contributions
Michelle C Ward, Conceptualization, Formal analysis, Investigation, Visualization, Methodology, Writing - original draft, Project administration; Nicholas E Banovich, Conceptualization, Formal analysis, Investigation, Methodology, Writing - review and editing; Abhishek Sarkar, Formal analysis, Methodology, Writing - review and editing; Matthew Stephens, Supervision; Yoav Gilad, Conceptualization, Supervision, Funding acquisition, Writing - review and editing

## Author ORCIDs
Michelle C Ward https://orcid.org/0000-0003-1485-320X
Nicholas E Banovich https://orcid.org/0000-0003-2604-3247
Abhishek Sarkar https://orcid.org/0000-0002-4636-9255
Yoav Gilad http://orcid.org/0000-0001-8284-8926

## Decision letter and Author response
Decision letter https://doi.org/10.7554/eLife.57345.sa1
Author response https://doi.org/10.7554/eLife.57345.sa2

# Additional files

## Supplementary files
• Supplementary file 1. Document containing tables listing experimental batches, cardiomyocyte purity, oxygen levels, RIN scores, sequencing read numbers, identified response genes, eGenes in A, B, C, D, dynamic eGenes, DARs, caQTLs in A, B, C, D, dynamic caQTLs, DMCpGs and GWAS trait overlaps.

• Supplementary file 2. Document containing the output of the CHT test to identify eQTLs in each condition. Values associated with the most significant SNP-gene association are included.

• Transparent reporting form

## Data availability
Sequencing data have been deposited in GEO under accession codes GSE144426.

The following dataset was generated:

| Author(s) | Year | Dataset title | Dataset URL | Database and Identifier |
|---|---|---|---|---|
| Ward MC, Banovich NE, Sarkar A, Stephens M, Gilad Y | 2021 | Dynamic effects of genetic variation on gene expression revealed following hypoxic stress in cardiomyocytes | https://www.ncbi.nlm.nih.gov/geo/query/acc.cgi?acc=GSE144426 | NCBI Gene Expression Omnibus, GSE144426 |

The following previously published datasets were used:

| Author(s) | Year | Dataset title | Dataset URL | Database and Identifier |
|---|---|---|---|---|
| GTEx Consortium | 2020 | GTEx eQTLs | https://www.gtexportal.org/home/ | GTEx portal V8 release, V8 |
| ENCODE Consortium | 2012 | Histone ChIP-seq | https://www.encodeproject.org | ENCODE, ENCBS684IAD |

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
