## [Decision Letter]

**Acceptance summary:**

This manuscript describes an in vitro system to study inter-individual variation of transcriptional and epigenetic responses to hypoxia using iPSC derived cardiomyocytes. Changes in gene expression are found without changes in chromatin accessibility or methylation, providing insight into key questions about the necessity of epigenetic changes for changes in gene expression. It also identifies regions of the genome responsible for context dependent-changes in gene expression.

**Decision letter after peer review:**

Thank you for submitting your article "Dynamic effects of genetic variation on gene expression revealed following hypoxic stress in cardiomyocytes" for consideration by *eLife*. Your article has been reviewed by two peer reviewers, and the evaluation has been overseen by a Reviewing Editor and Patricia Wittkopp as the Senior Editor. The reviewers have opted to remain anonymous.

The reviewers have discussed the reviews with one another and the Senior Editor has drafted this decision to help you prepare a revised submission.

Both reviewers agree that this manuscript is in principle appropriate for publication in *eLife* without any further experimentation; however, they both also raise a number of concerns about the data analysis and presentation that would need to be addressed satisfactorily before we can consider this work further. Although the typical practice for *eLife* is to provide authors with a single consolidated review, I felt it would be more beneficial in this case to provide the two reviews as is. They are generally consistent and both explain why they recommend the changes they do in detail.

Reviewer #2:

1) The authors present the range of cardiomyocyte purity. However, I think it would be helpful for this data to be presented in a table. Further, how do the purities correlate between the individual at different time points collected? Could some of the eQTLs identified really be a result of the variation in purity of the differentiations? Investigation and results regarding this possibility would be helpful if included in the manuscript.

2) The inferences made in this paper about the identification of dynamic eQTLs (and their absence) for a given loci rests on the assumption that the power to detect an eQTL is the same in every cell-type by condition. This is addressed in subsection “Dynamic eQTLs are revealed following hypoxia”, where it is clearly show (and acknowledged) that this is not the case. To avoid this problem the authors chose a lenient secondary cut-off at the point where p-values deviate from uniformity. Firstly, it would be helpful to formally test for deviation of uniformity here (rather than appearance on a graph). More importantly, utilising the same p-value threshold when the distribution of p-values is different will lead to variation in the false positive / false negative rates between conditions. This would lead to bias in the identification of dynamic eQTLs, as variation in eQTL detection is a function of the FP/FN rates. Can this be addressed, and it clearly demonstrated that the comparison between conditions is based on the same ability to detect a true eQTL?

3) In subsection “Dynamic eQTLs are revealed following hypoxia”, the authors state that the genes that were not eQTLs in any condition were uniform. However, the figures suggest significant skewing to more significant p-values. Please test for the uniform distribution with a chi-squared test. The authors suggest that they would expect the eQTLs identified in one condition to be uniform in another condition. This relies a the huge assumption that the largest contributing factor to significant eQTLs in each condition is the condition itself, not the cell type and that there will not be a significant number of eQTLs that are consistent across all conditions. However, the authors have provided no evidence to support this claim or this assumption. Please provide the requisite evidence to support these claims and assumptions.

4) Related to this, in paragraph three, it is indicated that based on the FDR calculations, half of the dynamic eQTLs that they identified are not truly dynamic. However, given this is the main conclusion of the paper, it really would be beneficial if some additional work could be done to ascertain the true from false dynamic eQTLs. Particularly if this work is going to be used for any translational research. One option would be to create a null distribution of the effect sizes for genes in each condition and test for enrichment of the "dynamic eQTLs" across the different conditions to test demonstrate whether they are actually identifying truly dynamic eQTLs. Further, I think different FDR cutoffs would have been more appropriate for identifying truly dynamic eQTLs since that is the main focus of the manuscript.

5) The analysis of the GTEx and GWAS overlap is unclear and would benefit from some more careful explanations. Reading this section, I had the following concerns:

i) How were the 14 tissue types randomly selected? And why not include all GTEx tissue types?

ii) How was overlap between dynamic eQTL and GTEx eQTL determined? Many eGenes are regulated by different eQTL – was the overlap based ona. Rank test of p-values between datasets for example? That would help provide confidence it is a true shared effect rather than unlinked eQTL.

iii) Are the allelic effects in the same direction?

iv) Why perform two stage comparison between eQTL and GWAS variants from all traits, then cardiac traits? Was multiple testing corrected for?

v) How do you rule out whether the overlap is just by chance, pleiotropy, linkage, or a causal relationship? These different scenarios have significant impact on the conclusions that are drawn. As it currently stands, it is implied that the overlap is a shared causal relationship.

6) One of the premises of this work is the statement made at the end of paragraph one of the Introduction, where the authors conclude that because the majority GTEx eQTLs are shared across tissues, they probably do not contribute to disease in tissue-specific manners. However, this statement makes two significant assumptions that are not addressed. 1) They assume that the downstream effects and signalling are the same regardless of cell type given they have the same eQTL. However, because different genes and pathways are activated in different tissues, this assumption may not hold true across all tissues. 2) They ignore the ascertainment bias associated with large additive cis-eQTL (which are more likely to overlap between tissues). The mean proportion of expression h2 that can be found to overlap between tissues is 0.02 (2%) in GTEX. The assumption that the remaining 98% are shared between tissues isn't supported by evidence from trans-eQTL mapping or single cell eQTL analyses. In my opinion, clarifying this statement in the Introduction as justification for the work would be beneficial.

7) I found that manuscript in general to be difficult to read and easily interpret the analysis conducted, and conclusions based on specific results. I would suggest a careful eye on editing a review would be very helpful. But here are a few points that I noted specifically

i) The notation used for supplementary figures and tables presented as "Figure X – Supplement Y". It would help to refer to the supplementary figures as "Supplementary Figure X" instead.

ii) Subsection “Dynamic eGenes are enriched for response genes and transcription factors” – it is not always entirely clear whether the authors are referring to transcription factor genes or transcription factors regulating the genes. You could include the word "gene" following "transcription factor" throughout the paragraph.

iii) Defining clearly what response genes and why they are important.

iv) Subsection “Potential limitations of our model” states that the power analysis indicates a 6% power to detect 38% heritability. Firstly, I assume, but it is not clear, that the test is against H0: h2=0? I understand the maths behind this, but that seems a really odd way of explaining power. Why not give a power curve?

v) How are the genomic features tested for enrichment of DARs?

vi) Subsection “Genomic features associated with differentially accessible regions (DARs)” paragraph two and Discussion paragraph four are confusing and unclear. It is hard to understand what the has been done.

vii) Throughout the manuscript, it is difficult to identify which timepoint were used for each assay. Please clarify this when introducing a new assay. Altering Figure 1 to make this clear graphically would help as well.

8) Some of the RIN numbers in RIN table are very low (i.e. 0 for 18852 in condition D). It's not clear if these samples were included in the analyses, but if they were some justification is needed. I worry about impact on the analysis, as essentially it would induce bias in those samples.

9) The section "DNA methylation levels are largely stable following hypoxia and re-oxygenation" is confusing and difficult to read. It is first stated that no DMCpGs were identified and then later stated that they were identified.

10) In subsection “Chromatin accessibility changes following hypoxia and re-oxygenation”, it is stated that ATAC-seq reads were selected that did not show allelic mapping biases. However, later testing for differential accessible regions. This filtering step would remove the ability to effectively identify differentially accessible regions with this step. Could you please either explain the reasoning for removing allelic mapping biases at this step, and/or explain why it would not impact differential accessibility analyses downstream.

11) It is concluded that the data indicate that the "underlying DNA sequence features can affect their epigenetic profile". However, I do not think that the results demonstrate this. They only demonstrate differential methylation in differentially accessible regions which are themselves not associated with genetic variants. I would suggest altering this sentence by removing "that the underlying DNA sequence features of these regions can affect their epigenetic profile"

Reviewer #3:

The authors established an in vitro system to study inter-individual variation of transcriptional and epigenetic responses to hypoxia using iPSC derived cardiomyocytes. Despite the modest sample size, they discovered more than a thousand eQTLs in total, of which 367 are dynamic eQTLs modulating the transcriptional response to hypoxia with implications for disease susceptibility.

Although the analyses demonstrated in the manuscript are solid and the results are reliable, authors failed to present their results effectively. I would recommend the authors to rewrite the paper to increase the accuracy and readability in general (as I suggested below).

In addition, I have one particular concern regarding the following important (but vague) claim presented in this paper, that is the transcriptional response to hypoxia is not mediated through epigenetic changes. Although the authors repeated the claim multiple times throughout the paper, it is hugely confounded by the power to detect epigenetic variations between conditions/individuals (i.e., DARs, caQTLs and DMCpGs). If the authors would keep the claim, they need to address the power issue appropriately.

Here are my specific comments:

1) Figure 1A should explain the overview of the experiment in more detail. There are three replicates for the three of 15 donors (mentioned in the main text) which are hidden in the panel. It is also nice to combine panel A and D in one panel to illustrate the whole experiment. According to the main text, one donor was dropped in ATAC-seq and two donors were dropped in methylation data which should also be shown in the panel.

2) It would be nice to summarise the 4 x 21 RNA-seq samples in the main figure. The authors used a linear mixed model for differential expression (DE) analysis. I'm wondering if the model can be used to perform a variance components analysis to decompose the total variance (for each gene) into “condition”, “individual”, “replication”, and residual variances (where the condition and replication are also treated as random effects). The estimated variance parameters across all genes are then summarised by a boxplot or similar where the residual variance is standardised as 1.0. The authors could repeat the analysis for ATAC-seq and methylation data to compare the difference of explained variances between molecular phenotypes. The analysis will provide clear interpretation on why detectable DARs, caQTLs and DMCpGs are so few. The similar analysis was performed e.g., in (Kilpinen et al., 2017).

3) Figure 2H is not so informative. Instead, the authors could extend the heatmap in Figure 2F to show how much of eGenes are classified into baseline eGenes, dynamic eGenes and shared eGenes by showing all genes with eQTLs at least in one condition. The authors could also add additional columns next to the heatmap showing which eGenes are response genes and/or transcription factors to support the enrichment analysis in the section “Dynamic eGenes are enriched for response genes and…”.

4) The DE analysis result is frequently used in the manuscript, but there is no summary data shown in the main text (e.g., volcano plots, DE genes shared/not shared with different conditions, etc.). It would be ideal to have an independent figure for DE analysis. Currently, Figure 2A-C are the result of DE analysis which are inconsistent with the figure title (i.e., “Hundreds of dynamic eQTLs are revealed…”).

5) How many dynamic eGenes are overlapping with the baseline eGenes? The Chi-square test performed in the section “Dynamic eGenes are enriched for response genes and transcription factors” seems inappropriate because the authors double-count the same genes in baseline eGenes as the background distribution. They need to subtract dynamic eGenes from the baseline eGenes to perform a proper Chi-square test.

6) The authors performed the enrichment analysis of transcription factors (TFs) in dynamic eGenes which could be confounded by response genes, because dynamic eGenes are likely to be response genes. In addition, even if dynamic eGenes are enriched with TFs, what is the biological interpretation? Why do regulatory variants affect TF dynamic eGenes more than non-TF dynamic eGenes?

7) It is often the case that the main figures are used to show anecdotal examples of biological statements raised in the main text and important data/results are hidden in the supplementary figures. The main figures must be used to support the author's claims in general (without cherry-picking an example gene). For example, the authors stated “most DARs have returned to baseline levels of accessibility by the first re-oxygenation condition.” with just numbers of DARs and the FOXO1 example which do not strongly support the claim. The authors should perform and highlight global analyses to support each claim from different angles. In fact, the claim was partly supported by the supplementary figure 3 (supplement 4A-C). Therefore, the authors first consider rearranging the figures to better support each claim made in the main text. Then, they can clarify which analysis is needed to make a strong statement.

8) The statement “These results suggests that gene expression changes…” was supported only by a limited number of DARs and not totally addressed yet. Recently, the “gene activity score” was proposed in the single-cell ATAC-seq analysis to overcome the sparseness of chromatin data (Granja et al., 2020). I'm wondering if it improves the power to detect DARs in the paper. This quantification also allows the authors to easily colocalise DARs with eGenes/response genes to prove (or disprove) the statement.

9) It is very surprising that CHT found only tens of caQTLs, while it mapped more than a thousand eQTLs. The authors should comment on that. If the authors use only variants overlapping with the accessibility region, does the power to detect caQTL increase? It seems 25Kb cis-region is too broad.

10) The statement “Changes in DNA methylation are therefore…” is hugely confounded by the power to detect DMCpGs with the higher noise rate in DNA methylation arrays. It is not conclusive with the modest sample size.

11) The authors should perform a colocalisation analysis (Giambartolomei et al., 2014) if GWAS summary statistics are available. If not, the authors are still able to calculate linkage disequilibrium (LD) between the GWAS lead variant and eQTL lead variant from CHT to make sure the GWAS locus and the eQTL are reasonably colocalised. The authors should show locuszoom plots in conjunction with boxplots in Figure 6 to demonstrate both the magnitude of statistical significance of eQTLs and LD (r^2^ index) with GWAS lead variants.

[Editors' note: further revisions were suggested prior to acceptance, as described below.]

Thank you for submitting your article "Dynamic effects of genetic variation on gene expression revealed following hypoxic stress in cardiomyocytes" for consideration by *eLife*. Your article has been reviewed by two peer reviewers, and the evaluation has been overseen by a Reviewing Editor and Patricia Wittkopp as the Senior Editor. The reviewers have opted to remain anonymous.

The reviewers have discussed the reviews with one another and the Reviewing Editor has drafted this decision to help you prepare a revised submission.

As stated by the reviewers in response to the first submission, the paper presents a timely, interesting and relevant advance. The initial comments raised have been largely addressed, however both reviewers continue raise concerns regarding the eQTL and GWAS overlap / colocalization analysis. We agree that these concerns are fundamental and should be addressed.

First, concerning the overlap with GTEX (R2), it seems entirely doable to perform this analysis on the level of individual variants, which also permit assessing consistency of effect directions, etc.

Second, both reviewers comment on the GWAS overlap/colocalization analysis. While we acknowledge that the resolution to definitely identify colocalization events will be limited, the authors should extend their analysis and/or town down the conclusions. The approach proposed by reviewer 3 seems sensible and viable, at least for a selected set of loci.

For reference, we have also included the complete reviewer reports.

Reviewer #2:

The authors have addressed most of the comments raised, although there are a couple of points that I feel would be beneficial to investigate further.

1) GTEX overlap – in the original review, a suggestion of testing for concordance of allelic effect direction be shown for the replication of the eQTL identified in this work and GTEX tissues. It is unclear what “analysis at the level of the gene” exactly means, but given that many genes are controlled by independent eQTL (loci) in difference tissues/cell types, overlapping an observation of eGenes doesn't allow conclusions of shared eQTL to be determined. Testing for the concordance of allelic effects for eSNPs identified from your analysis with GTEX eSNP results would provide much stronger evidence to support this conclusion.

2) GWAS overlap – "We next chose to analyze the data using a less stringent gene-based approach using GWAS traits of interest. The second analysis allows for the identification of potentially interesting regions where the causal SNP may be different in the eQTL and GWAS data" I am not sure what this means biologically and what conclusions can be drawn from this. What does it mean if there is no overlap in the SNPs between a eQTL and GWAS loci but that they are both in the same gene region? Is there an enrichment over what would be expected by chance? the statements "We did not mean to imply that the overlap is a shared causal relationship. We have changed the language to reflect this. Our goal is to illustrate that the dynamic eQTLs we identified may play a role in higher-level phenotypes." These sentences seems in opposition to one another.

Reviewer #3:

The authors have now addressed most of my questions and concerns. Yet, the result of GWAS colocalisation is still in question. I have previously suggested to perform a proper colocalisation analysis using COLOC package or to report the linkage disequilibrium index (r^2^ value) between GWAS index variant and eQTL lead variant. Unfortunately the authors just provided only P-values for associations which does not support the GWAS locus and eQTL are colocalised. The authors have to expand the cis-window (up to 1Mb) for a handful number of genes overlapping with GWAS loci to perform a proper colocalisation analsysis. It is crucial to provide the posterior probability of colocalisation for such a small number of GWAS overlapping genes.

---

## [Author Response]

Reviewer #2:1) The authors present the range of cardiomyocyte purity. However, I think it would be helpful for this data to be presented in a table. Further, how do the purities correlate between the individual at different time points collected? Could some of the eQTLs identified really be a result of the variation in purity of the differentiations? Investigation and results regarding this possibility would be helpful if included in the manuscript.

Please see Supplementary file 1 as indicated in the text for the purity of each cell line.

We did not measure purity at different time points, only at day 30 +/- 1 day, before the treatment on day 31-32. Because we split treatment cultures from a single differentiation culture, and all experiments were concluded within 48h after purity was determined by flow cytometry, it is unlikely that there are purity differences between conditions.

We have made this clearer in the Results and we have added a schematic of the experimental workflow as Figure 1—figure supplement 1. We have also amended Figure 2—figure supplement 4 to include cardiomyocyte marker gene expression levels across all conditions instead of just condition A as a representative condition. This data illustrates that there is no systematic bias in cardiomyocyte marker gene expression levels across conditions suggesting that the purity is indeed similar across conditions.

2) The inferences made in this paper about the identification of dynamic eQTLs (and their absence) for a given loci rests on the assumption that the power to detect an eQTL is the same in every cell-type by condition. This is addressed in subsection “Dynamic eQTLs are revealed following hypoxia”, where it is clearly show (and acknowledged) that this is not the case. To avoid this problem the authors chose a lenient secondary cut-off at the point where p-values deviate from uniformity. Firstly, it would be helpful to formally test for deviation of uniformity here (rather than appearance on a graph). More importantly, utilising the same p-value threshold when the distribution of p-values is different will lead to variation in the false positive / false negative rates between conditions. This would lead to bias in the identification of dynamic eQTLs, as variation in eQTL detection is a function of the FP/FN rates. Can this be addressed, and it clearly demonstrated that the comparison between conditions is based on the same ability to detect a true eQTL?

We tested for deviation from uniform distributions and added that data to the figure legend of Figure 3—figure supplement 2A. We believe that the assumption that power to detect eQTLs is similar in all conditions is a reasonable one as we now note because: (i) the cells originated from the same differentiation culture for each individual so the only factor distinguishing these samples should be the condition. (ii) We have the same number of individuals in each condition. (iii) RNA was extracted from all time points of the same individual at the same time (Supplementary file 1). (iv) RIN scores are similar across conditions (Supplementary file 1). (v) Sequencing read counts are similar across conditions (Figure 2—figure supplement 1). Moreover, the distribution of eQTL effect sizes across conditions is similar (new supplementary figure: Figure 3—figure supplement 1), and the values are correlated, further suggesting we have similar power to detect eQTLs across conditions.

The reviewer is correct that reliance on an arbitrary P-value when the distributions are different will lead to a different FDR. For that reason, our first – more stringent – cutoff is based on an FDR value, not a P-value. The secondary cutoff is based on a nominal P-value, which was determined based on visualization of the distributions. Importantly, we provide the FDR associated with different secondary P-value cutoffs in reviewer comment four below, thereby addressing the reviewer’s concern.

3) In subsection “Dynamic eQTLs are revealed following hypoxia”, the authors state that the genes that were not eQTLs in any condition were uniform. However, the figures suggest significant skewing to more significant p-values. Please test for the uniform distribution with a chi-squared test. The authors suggest that they would expect the eQTLs identified in one condition to be uniform in another condition. This relies a the huge assumption that the largest contributing factor to significant eQTLs in each condition is the condition itself, not the cell type and that there will not be a significant number of eQTLs that are consistent across all conditions. However, the authors have provided no evidence to support this claim or this assumption. Please provide the requisite evidence to support these claims and assumptions.

The experiments were concluded within 48 hours of testing a representative sample from each individual for purity. It does not seem reasonable that during that time the composition of the cultures has changed, certainly not in a way that will create a biased cell composition association with a specific treatment condition. We now acknowledge in the text that our conclusion relies on this assumption, and explain why we believe that it is quite a reasonable assumption to make.

We have tested for deviation from a uniform distribution and indicated the statistics in the figure legend of Figure 3—figure supplement 2B, and in the main text. However, given that the outcome of a chi-square test (rejection/non-rejection) could depend on how the data are binned, and it is not obvious how to do so in this case, we used the Kolmogorov–Smirnov test to determine whether our P-value distribution deviates from a uniform distribution as a simpler alternative.

4) Related to this, in paragraph three, it is indicated that based on the FDR calculations, half of the dynamic eQTLs that they identified are not truly dynamic. However, given this is the main conclusion of the paper, it really would be beneficial if some additional work could be done to ascertain the true from false dynamic eQTLs. Particularly if this work is going to be used for any translational research. One option would be to create a null distribution of the effect sizes for genes in each condition and test for enrichment of the "dynamic eQTLs" across the different conditions to test demonstrate whether they are actually identifying truly dynamic eQTLs. Further, I think different FDR cutoffs would have been more appropriate for identifying truly dynamic eQTLs since that is the main focus of the manuscript.

As the reviewer appreciates, it is not obvious how to distinguish between the true and false positives in this case. We supposed that one can use priors based on gene functions, but this approach seems prone to errors. We have now determined the number of dynamic eQTLs at different statistical cutoffs and their associated FDR (P=0.5: 117 dynamic eQTLs with 30% FDR; P=0.75: 53 eQTLs with 13% FDR; P=0.8: 37 eQTLs with 11% FDR;, P=0.9: 19 eQTLs with 7% FDR). We have added the results of one of these more stringent P-value cutoffs, where the FDR is quite low (7%) to the main text. We also now provide a supplementary figure highlighting six additional examples of dynamic eQTLs (Figure 3—figure supplement 3) indicating that our approach identifies dynamic eQTLs as expected.

5) The analysis of the GTEx and GWAS overlap is unclear and would benefit from some more careful explanations. Reading this section, I had the following concerns:i) How were the 14 tissue types randomly selected? And why not include all GTEx tissue types?

This number and the specific tissues were chosen in an arbitrary way. We were looking for a modest number of diverse tissue types to interrogate. This analysis illustrates that 98% of dynamic eGenes are eGenes in at least one of 14 tissues suggesting we are reaching saturation already. Adding more tissues therefore can’t add much more insight because we are already at 98% overlap. We clarified that the choice of tissues was arbitrary – effectively random in the text.

ii) How was overlap between dynamic eQTL and GTEx eQTL determined? Many eGenes are regulated by different eQTL – was the overlap based ona. Rank test of p-values between datasets for example? That would help provide confidence it is a true shared effect rather than unlinked eQTL.

As indicated in the Materials and methods this analysis was performed at a gene level; i.e. a gene’s expression level is known to be affected by genetic variation (eGene) in GTEx data. We asked how often do these GTEx eGenes overlap with our eGenes. We have made the approach we chose more explicit in the Results.

A more nuanced SNP-based analysis is actually quite challenging to perform and will require a different approach. As we are uncertain what the potential insight would be from such an analysis, on balance, we decided against it.

iii) Are the allelic effects in the same direction?

As the analysis we performed was based on a gene level we are unable to compare the genotype/allelic effects. There is overwhelming overlap between our findings and GTEx. While we agree that a more nuanced approach will provide more details, we argue that this is a minor point in the paper, and further analysis in this section will not substantially add insight.

iv) Why perform two stage comparison between eQTL and GWAS variants from all traits, then cardiac traits? Was multiple testing corrected for?

To identify genomic loci that may be relevant for complex phenotypes, we performed two independent analyses to intersect our eQTL data with data from GWAS studies. We chose two different analytical approaches to identify as many potentially interesting loci as possible. We first chose to analyze the data using an unbiased, stringent SNP-based approach using all available GWAS traits. We next chose to analyze the data using a less stringent gene-based approach using GWAS traits of interest. The second analysis allows for the identification of potentially interesting regions where the causal SNP may be different in the eQTL and GWAS data.

In both of these analyses we searched for direct overlaps of either SNPs or genes. We did not correct for multiple testing. In line with the comment below we have changed the language to more accurately reflect the analysis we performed.

We have also now included a locus-specific co-localization-based analysis as requested by reviewer three. We have clarified the choice of these three different approaches in the text.

v) How do you rule out whether the overlap is just by chance, pleiotropy, linkage, or a causal relationship? These different scenarios have significant impact on the conclusions that are drawn. As it currently stands, it is implied that the overlap is a shared causal relationship.

We did not mean to imply that the overlap is a shared causal relationship. We have changed the language to reflect this. Our goal is to illustrate that the dynamic eQTLs we identified may play a role in higher-level phenotypes.

6) One of the premises of this work is the statement made at the end of paragraph one of the Introduction, where the authors conclude that because the majority GTEx eQTLs are shared across tissues, they probably do not contribute to disease in tissue-specific manners. However, this statement makes two significant assumptions that are not addressed. 1) They assume that the downstream effects and signalling are the same regardless of cell type given they have the same eQTL. However, because different genes and pathways are activated in different tissues, this assumption may not hold true across all tissues. 2) They ignore the ascertainment bias associated with large additive cis-eQTL (which are more likely to overlap between tissues). The mean proportion of expression h2 that can be found to overlap between tissues is 0.02 (2%) in GTEX. The assumption that the remaining 98% are shared between tissues isn't supported by evidence from trans-eQTL mapping or single cell eQTL analyses. In my opinion, clarifying this statement in the Introduction as justification for the work would be beneficial.

As the reviewer notes this is a complex issue, which was overly simplified in our statement about the consequences of sharing of eQTLs across tissues. We have removed this sentence and instead include a statement about how a variety of approaches are necessary to identify the molecular basis of disease-associated loci.

7) I found that manuscript in general to be difficult to read and easily interpret the analysis conducted, and conclusions based on specific results. I would suggest a careful eye on editing a review would be very helpful. But here are a few points that I noted specifically

We have addressed each point below.

i) The notation used for supplementary figures and tables presented as "Figure X – Supplement Y". It would help to refer to the supplementary figures as "Supplementary Figure X" instead.

The figures have been referenced in the format specified by *eLife*.

ii) Subsection “Dynamic eGenes are enriched for response genes and transcription factors” – it is not always entirely clear whether the authors are referring to transcription factor genes or transcription factors regulating the genes. You could include the word "gene" following "transcription factor" throughout the paragraph.

By “transcription factor” we mean any gene that is annotated as a transcription factor by Lambert et al., not transcription factors that regulate a particular set of genes. This is indicated in the Materials and methods but we have made this more explicit in the Results.

iii) Defining clearly what response genes and why they are important.

We should have explicitly defined response genes. We have now done so in the text.

iv) Subsection “Potential limitations of our model” states that the power analysis indicates a 6% power to detect 38% heritability. Firstly, I assume, but it is not clear, that the test is against H0: h2=0? I understand the maths behind this, but that seems a really odd way of explaining power. Why not give a power curve?

We clarify that the test is against H_0_: λ = 0, where λ is the standardized effect size.

This corresponds to the typical regression-based approach to map QTLs, assuming that genotypes and phenotypes have been standardized.

We plotted the power curve as a function of standardized effect size, fixing the sample size to the size of the current study, in Figure 7—figure supplement 3 (green curve).

Assuming a single causal variant, the standardized effect size be used to derive the heritability explained by that variant.

We reported power to detect an effect of a given true effect size, holding the sample size fixed to the size of the current study. We gave the true effect size in terms of heritability explained, rather than giving the standardized effect size, in order to make the results of the power analysis easier to interpret. The standardized effect size does not correspond to log fold change, which would be an alternative, interpretable choice of units.

In particular, the median eQTL heritability over all genes is h^2^ = 0.16, as previously reported by Gusev et al., 2016 and Wheeler et al., 2016. This reference point makes clear the relative power of this study compared to the GTEx and DGN studies for eQTL mapping.

We have added the Gusev and Wheeler studies for reference to the Discussion.

v) How are the genomic features tested for enrichment of DARs?

We have added the statistical test used to identify promoter- and enhancer-associated marks (chi-squared) to the Results. We have also added more details in the Materials and methods to describe how MEME identifies enriched motifs.

vi) Subsection “Genomic features associated with differentially accessible regions (DARs)” paragraph two and Discussion paragraph four are confusing and unclear. It is hard to understand what the has been done.

We have re-written these sections to make them clearer.

vii) Throughout the manuscript, it is difficult to identify which timepoint were used for each assay. Please clarify this when introducing a new assay. Altering Figure 1 to make this clear graphically would help as well.

We used all time points in all assays and differences between all timepoints were tested in all assays.

We have clarified this in the text, re-designed Figures 1 and 2 and added a supplementary figure with the experimental setup (Figure 1—figure supplement 1) to illustrate this point.

8) Some of the RIN numbers in RIN table are very low (i.e. 0 for 18852 in condition D). It's not clear if these samples were included in the analyses, but if they were some justification is needed. I worry about impact on the analysis, as essentially it would induce bias in those samples.

The median RIN score across 84 samples is 9.1 (9.4 for A, 8.7 for B, 9.1 for C and 9.2 for D). All RIN scores are >8, except 18852A which is 6 and 18852D which is listed as 0. The RIN score for 18852D could not be calculated by the bioanalyzer software, despite the two ribosomal subunits evident in the sample. We should have listed this score as “not detected” not “0”. Given that we have multiple replicates for 18852, and that only one replicate can be used in the eQTL and differential expression analysis, we chose not to include this replicate in downstream analysis.

We have added clarification text to where we mention RIN scores in the Materials and methods, changed the RIN score of 18852D to “not detected” in the table to more accurately reflect this score, and plotted the RIN values in Figure 2—figure supplement 1A to highlight this data.

9) The section "DNA methylation levels are largely stable following hypoxia and re-oxygenation" is confusing and difficult to read. It is first stated that no DMCpGs were identified and then later stated that they were identified.

We did not find any differentially methylated CpGs when considering all CpGs. We did find a handful of differentially methylated CpGs when restricting our analysis to a smaller set of CpGs (those in CpG islands etc).

We have made this distinction, and the rationale behind it, clearer in the text. We also now do not use the term DMCpGs when considering all CpGs. We only introduce the term when we subset CpGs and actually identify differences between conditions. This distinction is also now illustrated in re-designed Figure 6.

10) In subsection “Chromatin accessibility changes following hypoxia and re-oxygenation”, it is stated that ATAC-seq reads were selected that did not show allelic mapping biases. However, later testing for differential accessible regions. This filtering step would remove the ability to effectively identify differentially accessible regions with this step. Could you please either explain the reasoning for removing allelic mapping biases at this step, and/or explain why it would not impact differential accessibility analyses downstream.

We treated the ATAC-seq data as we did the RNA-seq data i.e. used the WASP algorithm to reprocess mapped reads to reduce reference bias as described in the Materials and methods. This step does not impact the ability to identify differentially accessible regions across conditions, it merely removes ambiguous reads. We have clarified in the Results that we only maintain ATAC-seq reads that map unambiguously to the genome.

11) It is concluded that the data indicate that the "underlying DNA sequence features can affect their epigenetic profile". However, I do not think that the results demonstrate this. They only demonstrate differential methylation in differentially accessible regions which are themselves not associated with genetic variants. I would suggest altering this sentence by removing "that the underlying DNA sequence features of these regions can affect their epigenetic profile"

We have altered this sentence as suggested.

Reviewer #3:The authors established an in vitro system to study inter-individual variation of transcriptional and epigenetic responses to hypoxia using iPSC derived cardiomyocytes. Despite the modest sample size, they discovered more than a thousand eQTLs in total, of which 367 are dynamic eQTLs modulating the transcriptional response to hypoxia with implications for disease susceptibility.Although the analyses demonstrated in the manuscript are solid and the results are reliable, authors failed to present their results effectively. I would recommend the authors to rewrite the paper to increase the accuracy and readability in general (as I suggested below).In addition, I have one particular concern regarding the following important (but vague) claim presented in this paper, that is the transcriptional response to hypoxia is not mediated through epigenetic changes. Although the authors repeated the claim multiple times throughout the paper, it is hugely confounded by the power to detect epigenetic variations between conditions/individuals (i.e., DARs, caQTLs and DMCpGs). If the authors would keep the claim, they need to address the power issue appropriately.

We used 13-14 biological replicates (different individuals) in our study. Previous studies have generally used just 1-6 biological replicates to identify differences in chromatin profiles or DNA methylation levels under conditions of stress (Aref-Eshghi, Am J Physio, Cell Physiol, 2020; Marr, Plos Path., 2014; Pacis, GR, 2015). It is also our experience that when we measure both gene expression and DNA methylation changes between sample groups, the number of DNA methylation changes often far outnumber the gene expression changes (Burrows/Banovich, Plos Genetics, 2016; Natri/Bobowik, Plos Genetics, 2020). Lack of power is always a reasonable explanation, but compared to previous studies, our observations are quite surprising. Nevertheless, we have now softened our claims about the lack of differences we observed in the Results section, and removed some of the conclusions we drew.

Here are my specific comments:1) Figure 1A should explain the overview of the experiment in more detail. There are three replicates for the three of 15 donors (mentioned in the main text) which are hidden in the panel. It is also nice to combine panel A and D in one panel to illustrate the whole experiment. According to the main text, one donor was dropped in ATAC-seq and two donors were dropped in methylation data which should also be shown in the panel.

We have re-designed Figure 1 to combine A and D and now indicate exactly which samples were collected.

2) It would be nice to summarise the 4 x 21 RNA-seq samples in the main figure. The authors used a linear mixed model for differential expression (DE) analysis. I'm wondering if the model can be used to perform a variance components analysis to decompose the total variance (for each gene) into “condition”, “individual”, “replication”, and residual variances (where the condition and replication are also treated as random effects). The estimated variance parameters across all genes are then summarised by a boxplot or similar where the residual variance is standardised as 1.0. The authors could repeat the analysis for ATAC-seq and methylation data to compare the difference of explained variances between molecular phenotypes. The analysis will provide clear interpretation on why detectable DARs, caQTLs and DMCpGs are so few. The similar analysis was performed e.g., in (Kilpinen et al., 2017).

We have now created an extra main text figure where we show the differential expression analysis (Figure 2). These results are largely in line with the findings from our previous work (Ward and Gilad, 2019), which is why we had not initially highlighted them in this paper.

Using a linear mixed model, we calculated the proportion of variance explained by individual, condition, replicate (we only have replicate data for the RNA-seq and DNA methylation data), and the residual variance. A boxplot of variance components for all three phenotypes is now included as Figure 6—figure supplement 4. We now note in the main text that we observe a lower proportion of variance explained by “condition” in the DNA methylation data (0.8%) and chromatin accessibility data (3%) compared to the gene expression data (6%, t-test, *P* < 2x10^-16^), and a corresponding increase in the residual variance. Comparing the gene expression data with the DNA methylation data in particular, reveals that while there is approximately an order of magnitude lower variance in the “condition” component in the DNA methylation data, there is a similar proportion of variance explained by “individual” (44% for gene expression, 40% for DNA methylation, and 28% for chromatin accessibility). These results suggest that noise alone cannot explain the relatively small number of chromatin accessibility and DNA methylation changes.

In terms of the low number of caQTLs, we specified in the Results section that we restricted tested SNPs to those within the 128,672 open chromatin regions we identified. We chose to do this so that any effects would be interpretable i.e. a SNP affects a transcription factor binding site, altering accessibility of the region. This means we were testing many fewer variants than the eQTL analyses. We may well be underpowered to detect these effects in our modest sample size. We have added text to the manuscript to make this point more explicit.

3) Figure 2H is not so informative. Instead, the authors could extend the heatmap in Figure 2F to show how much of eGenes are classified into baseline eGenes, dynamic eGenes and shared eGenes by showing all genes with eQTLs at least in one condition. The authors could also add additional columns next to the heatmap showing which eGenes are response genes and/or transcription factors to support the enrichment analysis in the section “Dynamic eGenes are enriched for response genes and…”.

We appreciate the suggestion on how to make this figure more informative. Unfortunately this didn’t work in practice partly because of the distinction between eSNP-based and eGene-based analyses and partly because of the available software. Instead, to address the comment we have now made a clear distinction between SNP-based results and eGene-based results, added an extra panel with a heatmap which includes all eQTLs identified in all conditions to provide a more global view of the data and to show how much “sharing” there is between conditions (Figure 3B), and re-designed a panel to include the response gene and TF data (Figure 3F).

4) The DE analysis result is frequently used in the manuscript, but there is no summary data shown in the main text (e.g., volcano plots, DE genes shared/not shared with different conditions, etc.). It would be ideal to have an independent figure for DE analysis. Currently, Figure 2A-C are the result of DE analysis which are inconsistent with the figure title (i.e., “Hundreds of dynamic eQTLs are revealed…”).

We have made a new Figure 2 which displays pairwise and joint tests in the RNA-seq data using panels from the original Figure 2 and supplement.

5) How many dynamic eGenes are overlapping with the baseline eGenes? The Chi-square test performed in the section “Dynamic eGenes are enriched for response genes and transcription factors” seems inappropriate because the authors double-count the same genes in baseline eGenes as the background distribution. They need to subtract dynamic eGenes from the baseline eGenes to perform a proper Chi-square test.

This was an error on our part. As the reviewer noted a subset of dynamic eGenes are present at baseline – these are the 37 suppressed eQTLs. We have now excluded these genes to compare only baseline eGenes and induced eGenes so that this is now a mutually exclusive set. This in fact increases the enrichment of both response genes and transcription factors, which we now report in the text.

6) The authors performed the enrichment analysis of transcription factors (TFs) in dynamic eGenes which could be confounded by response genes, because dynamic eGenes are likely to be response genes. In addition, even if dynamic eGenes are enriched with TFs, what is the biological interpretation? Why do regulatory variants affect TF dynamic eGenes more than non-TF dynamic eGenes?

We have now investigated the enrichment of TFs in induced eGenes that are response genes compared to induced eGenes that are non-response genes. TFs amongst induced eGenes are more likely to be response genes than non-response genes (P = 0.02); however there is no difference in the proportion of response genes between TFs amongst baseline eGenes and TFs amongst induced eGenes (38% vs. 39%). We have added these results to the main text.

The enrichment we observe in TFs amongst our induced eGenes suggests that latent genetic variation can have multiple downstream effects on gene expression including gene targets of TF genes. This could provide a mechanism for the appearance of induced eGenes that are neither response genes nor TF genes. We have added the implication of our findings to the main text.

7) It is often the case that the main figures are used to show anecdotal examples of biological statements raised in the main text and important data/results are hidden in the supplementary figures. The main figures must be used to support the author's claims in general (without cherry-picking an example gene). For example, the authors stated “most DARs have returned to baseline levels of accessibility by the first re-oxygenation condition.” with just numbers of DARs and the FOXO1 example which do not strongly support the claim. The authors should perform and highlight global analyses to support each claim from different angles. In fact, the claim was partly supported by the supplementary figure 3 (supplement 4A-C). Therefore, the authors first consider rearranging the figures to better support each claim made in the main text. Then, they can clarify which analysis is needed to make a strong statement.

We have re-designed Figure 4 and 6 to include the genome-wide analyses performed for the ATAC-seq and DNA methylation data that were previously shown in the supplementary figures, and we have edited the text accordingly.

8) The statement “These results suggests that gene expression changes…” was supported only by a limited number of DARs and not totally addressed yet. Recently, the “gene activity score” was proposed in the single-cell ATAC-seq analysis to overcome the sparseness of chromatin data (Granja et al., 2020). I'm wondering if it improves the power to detect DARs in the paper. This quantification also allows the authors to easily colocalise DARs with eGenes/response genes to prove (or disprove) the statement.

The referenced statement refers to the identification of caQTLs not DARs. However, given the other reviewer comments we have removed this particular statement and softened the language in this section. The referenced gene activity score is for aggregating regulatory regions across single cells. We aggregate open chromatin regions across individuals within each condition (by merging peaks of read enrichment) and then look for differences in read count between conditions and individuals. It is not clear to us how this tool would work with our bulk data, or how it would improve power to detect differences between conditions.

9) It is very surprising that CHT found only tens of caQTLs, while it mapped more than a thousand eQTLs. The authors should comment on that. If the authors use only variants overlapping with the accessibility region, does the power to detect caQTL increase? It seems 25Kb cis-region is too broad.

As we stated in the Results and in the Materials and methods, we tested for caQTLs as we did for eQTLs except we did not use a 25 kb window, we restricted SNPs to those within the 128,672 open chromatin regions.

10) The statement “Changes in DNA methylation are therefore…” is hugely confounded by the power to detect DMCpGs with the higher noise rate in DNA methylation arrays. It is not conclusive with the modest sample size.

We would argue that most studies aimed at identifying differences between two conditions do not have 13 biological replicates. Nevertheless, we have removed this statement.

11) The authors should perform a colocalisation analysis (Giambartolomei et al., 2014) if GWAS summary statistics are available. If not, the authors are still able to calculate linkage disequilibrium (LD) between the GWAS lead variant and eQTL lead variant from CHT to make sure the GWAS locus and the eQTL are reasonably colocalised. The authors should show locuszoom plots in conjunction with boxplots in Figure 6 to demonstrate both the magnitude of statistical significance of eQTLs and LD (r^2^ index) with GWAS lead variants.

We only have a small number of overlapping results here, so we chose an alternative approach: We were able to obtain full summary statistics from the CARDIoGRAMplusC4D Consortium for coronary artery disease (CAD) and myocardial infarction (MI). From both significant and dynamic eQTLs we tested the lead eSNP against the full GWAS results and identified two new associations. The first is an association between a dynamic eQTL (regulating *EIF2B2*) and CAD. The p-value of the dynamic eSNP is 4.7x10^-5^ in the GWAS and the most significant SNP in the GWAS peak has a p-value of 9.9x10^-8^. As we tested a relatively narrow window around genes in our eQTL analysis, our summary statistics do not overlap the full peak. We also identified an association between a significant eQTL in condition B that is significantly associated with MI (*CARM1*). We have added both of these loci to the manuscript, and now highlight *EIF2B2* with locus zoom plots in Figure 7 and *CARM1* in Figure 7—figure supplement 2.

The two original loci that we highlighted were *RNF16*6, associated with varicose veins, and *ZC3HC1*, associated with MI. Varicose vein GWAS summary statistics were only available for the significant associations. We include a locus zoom plot of the *RNF166* region with only the significant GWAS hits (Author response image 1). The GWAS region around *ZC3HC1* does not show a clear peak despite being a known MI locus (Author response image 1). We felt that neither of these locus zoom plots are striking as examples to illustrate in the main paper. However, we believe that these are nevertheless potentially two interesting loci worth mentioning. We have moved the original *RNF166* and *ZC3HC1* loci eQTL boxplots to Figure 7—figure supplement 1.

[Editors' note: further revisions were suggested prior to acceptance, as described below.]

Reviewer #2:The authors have addressed most of the comments raised, although there are a couple of points that I feel would be beneficial to investigate further.1) GTEX overlap – in the original review, a suggestion of testing for concordance of allelic effect direction be shown for the replication of the eQTL identified in this work and GTEX tissues. It is unclear what “analysis at the level of the gene” exactly means, but given that many genes are controlled by independent eQTL (loci) in difference tissues/cell types, overlapping an observation of eGenes doesn't allow conclusions of shared eQTL to be determined. Testing for the concordance of allelic effects for eSNPs identified from your analysis with GTEX eSNP results would provide much stronger evidence to support this conclusion.

We have now performed the GTEx overlap analysis of dynamic and shared eQTLs at the level of eSNP-eGene pairs rather than eGenes. 326 of 367 dynamic eSNPs are tested in GTEx. 32 dynamic eSNP-eGene pairs are eQTLs in left ventricle heart tissue (9.8%), and 140 are eQTLs in at least one of the other 49 tissues assayed by GTEx (42.9%). For the shared eQTL analysis we selected the 61 eGenes present in all conditions, identified the most significant eSNP in condition A and then determined whether this eSNP was significant in conditions B,C and D. 19 of 20 such shared eSNPs are tested in GTEx. 6 shared eSNP-eGene pairs are eQTLs in heart tissue (31.6%), and 6 are an eQTL in at least one other tissue (31.6%). In line with our previous eGene analysis, shared eSNPs are more likely to be eSNPs in heart tissue than dynamic eSNPs (Chi-squared test; p=0.01). To further investigate eQTL concordance, we compared the eQTL effect size of our dynamic and shared eQTLs to the eQTL effect size in heart tissue determined by the GTEx consortium. Shared eQTLs overlapping heart tissue eQTLs (n=6) have a higher concordance of effect than dynamic eQTLs overlapping heart tissue eQTLs (n=32; r2=0.73 vs. r2=0.12). We are only able to access the significant eQTLs identified by the GTEx consortium and were therefore unable to determine broader concordance across data sets.

We have replaced the GTEx eGene analysis with this new variant-based analysis in the Results section and Materials and methods section.

2) GWAS overlap – "We next chose to analyze the data using a less stringent gene-based approach using GWAS traits of interest. The second analysis allows for the identification of potentially interesting regions where the causal SNP may be different in the eQTL and GWAS data" I am not sure what this means biologically and what conclusions can be drawn from this. What does it mean if there is no overlap in the SNPs between a eQTL and GWAS loci but that they are both in the same gene region? Is there an enrichment over what would be expected by chance? the statements "We did not mean to imply that the overlap is a shared causal relationship. We have changed the language to reflect this. Our goal is to illustrate that the dynamic eQTLs we identified may play a role in higher-level phenotypes." These sentences seems in opposition to one another.

Our study population are the Yoruba from West Africa. Unfortunately, the vast majority of GWAS studies, including the CAD and MI studies reported here, are carried out in predominantly European populations which have an inherently different underlying LD structure. Thus, while our GWAS analysis includes a SNP based approach, we also felt that casting a wider net connecting eQTLs with genes which have orthogonal evidence of being involved in heart disease was prudent. Specifically, given the underlying differences in LD structure between European and African populations we anticipate that some variants identified by GWAS studies will have weak co-localization with eQTLs identified in our study, but the fact that these SNPs are acting to alter the expression of genes implicated in heart disease suggests they may be important for disease risk.

We have included the caveats of this analysis and explained our approach in the manuscript.

Reviewer #3:The authors have now addressed most of my questions and concerns. Yet, the result of GWAS colocalisation is still in question. I have previously suggested to perform a proper colocalisation analysis using COLOC package or to report the linkage disequilibrium index (r^2^ value) between GWAS index variant and eQTL lead variant. Unfortunately the authors just provided only P-values for associations which does not support the GWAS locus and eQTL are colocalised. The authors have to expand the cis-window (up to 1Mb) for a handful number of genes overlapping with GWAS loci to perform a proper colocalisation analsysis. It is crucial to provide the posterior probability of colocalisation for such a small number of GWAS overlapping genes.

We appreciate the comments from the reviewer and indeed, failed to include the requested LD analysis in the first revision of this paper. Unfortunately, performing a full colocalization analysis using an extended set of variants that overlap the lead GWAS variant is non-trivial as we cannot easily add new variants to the combined haplotype test without performing a near complete reanalysis of the data. However, we can and have carried out the suggested LD analysis. Using LDlink we tested for the r^2^ value between the lead GWAS variants near *EIF2B2* and *CARM1* and our eSNPs (in the YRI population). We found the lead GWAS variant associated with CAD risk located near the *EIF2B2* gene (rs3832966) was in high LD with our dynamic eSNP (rs12588981) associated with expression of the *EIF2B2* gene (r^2^ = 0.96, p < 0.001). Similarly, the lead GWAS variant associated with MI near the *CARM1* gene (rs4804142) is in lower, but still significant, LD with our condition B eSNP (rs8105092), which is associated with *CARM1* expression (r^2^ 0.32 p < 0.001).

We have added text to describe this analysis in the manuscript in the Results section, and the Materials and methods section. We have also added two supplementary figures of the LD matrix plots (Figure 7—figure supplement 2 and Figure 7—figure supplement 3D).

References:

Giambartolomei, Claudia, Damjan Vukcevic, Eric E. Schadt, Lude Franke, Aroon D. Hingorani, Chris Wallace, and Vincent Plagnol. 2014. "Bayesian Test for Colocalisation between Pairs of Genetic Association Studies Using Summary Statistics." PLoS Genetics 10 (5): e1004383.

Granja, Jeffrey M., M. Ryan Corces, Sarah E. Pierce, S. Tansu Bagdatli, Hani Choudhry, Howard Y. Chang, and William J. Greenleaf. 2020. "ArchR: An Integrative and Scalable Software Package for Single-Cell Chromatin Accessibility Analysis." bioRxiv. https://doi.org/10.1101/2020.04.28.066498.

Kilpinen, Helena, Angela Goncalves, Andreas Leha, Vackar Afzal, Kaur Alasoo, Sofie Ashford, Sendu Bala, et al. 2017. "Common Genetic Variation Drives Molecular Heterogeneity in Human iPSCs." Nature 546 (7658): 370-75.